# ML-based top taggers: Performance, uncertainty and impact of tower & tracker data integration

Rameswar Sahu[⋆] and Kirtiman Ghosh[†]

Homi Bhabha National Institute, Training School Complex,
Anushakti Nagar, Mumbai 400094, India
Institute of Physics, Bhubaneswar, Sachivalaya Marg,
Sainik School, Bhubaneswar 751005, India

⋆ rameswar.s@iopb.res.in , † kirti.gh@gmail.com

## Abstract

Machine learning algorithms have the capacity to discern intricate features directly from raw data. We demonstrated the performance of top taggers built upon three machine learning architectures: a BDT that uses jet-level variables (high-level features, HLF) as input, a CNN (a miniature version of ResNet) trained on the jet image, and a GNN (LorentzNet) trained on the particle cloud representation of a jet utilizing the 4-momentum (low-level features, LLF) of the jet constituents as input. We found significant performance enhancement for all three classes of classifiers when trained on combined data from calorimeter towers and tracker detectors. The high resolution of the tracking data not only improved the classifier performance in the high transverse momentum region, but the information about the distribution and composition of charged and neutral constituents of the fat jets and subjets helped identify the quark/gluon origin of subjets and hence enhances top tagging efficiency. The LLF-based classifiers, such as CNN and GNN, exhibit significantly better performance when compared to HLF-based classifiers like BDT, especially in the high transverse momentum region. Nevertheless, the LLF-based classifiers trained on constituents' 4-momentum data exhibit substantial dependency on the jet modeling within Monte Carlo generators. The composite classifiers, formed by stacking a BDT on top of a GNN/CNN, not only enhance the performance of LLF-based classifiers but also mitigate the uncertainties stemming from the showering and hadronization model of the event generator. We have conducted a comprehensive study on the influence of the fat jet's reconstruction and labeling procedure on the efficiency of the classifiers.

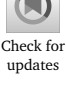

# 1 Introduction

Since its commencement, the Large Hadron Collider (LHC) [1] at CERN has been looking for evidence of physics beyond the Standard Model (BSM). While the discovery of the Higgs [2,3] is a remarkable success for the LHC experiment, it serves to reinforce solely the legitimacy of the Standard Model (SM). The absence of any solid evidence supporting BSM physics has motivated researchers to progressively explore higher energy scales. Such high energies facilitate the production of boosted heavy SM particles like the top quark, $W/Z$-boson, and the Higgs boson. The hadronic decays of the boosted SM particles lead to a collimated cluster of quarks, manifesting as large radius (large-$R$) single jets (fat jets) with distinctive features. At the LHC, the sub-structure features of fat jets resulting from the hadronic decay of boosted top quarks, W/Z-bosons or the Higgs boson have been widely utilized[1] to search for heavy BSM resonances within various new physics scenarios such as supersymmetry [6–8], extra-dimensional models,

---

[1]Considering the hadronically decaying boosted top quarks or W/Z bosons offers several advantages when designing search strategies for the heavy BSM resonances that decay into massive SM particles. On the one hand, the enhanced hadronic decay branching ratios of top quarks or $W/Z$ bosons lead to a higher signal rate. The hadronic decay products of top quarks or W/Z bosons being visible at the LHC detectors enables the kinematic reconstruction of the decay cascade for specific BSM resonances [4,5].

leptoquark models, different gauge and field extensions of the SM [4, 5, 9–11], etc. Efficient identification of the particle identity of the fat jet becomes essential to improve the sensitivity of the LHC and future colliders. This necessitates a substantial shift in the analysis strategy and demands the development of new and innovative methodologies for tagging the particle identities of the fat jets.

Traditional methods for jet-tagging rely on constructing high-level discriminants from the jet substructure information, the so-called jet substructure observables[2] [19–30]. The usefulness of jet substructure observables in distinguishing large-$R$ jets resulting from hadronically decaying boosted heavy SM particles over QCD quark/gluon jets has been demonstrated and widely accepted by experimental collaborations [31–36]. Incorporating machine learning-based techniques into the task of jet classification opens up new and unique directions. These methods leverage the fine granularity of the LHC detectors to construct highly specialized observables from the four-momentum of the constituents of the jet. Over the past decade, various neural network-based architectures have been developed [37–62] and have demonstrated substantial enhancement in the classification efficiencies compared to traditional substructure-based techniques. Apart from architectural complexity, these algorithms differ in the representation of the input dataset. While Linear classifiers [49] and BDTs [63, 64] are trained on jet-level observables constructed from the jet substructure information, Convolutional Neural Networks (CNNs) [40, 41, 44, 52, 54, 55, 65–72], Recurrent Neural Networks (RNNs) [71, 73–75], Graph Neural Networks (GNNs) [46, 57, 76–86], Recursive Neural Networks (RvNNs) [45, 61, 87, 88], Fisher's Linear Discriminant [56], Locally Connected Networks [53], etc. are directly trained on pure or transformed four-momentum data[3] of the jet constituent. On the other hand, Multi-Layer Perceptrons (MLPs) can be trained on both jet-level observable data [48, 89–92] as well as constituents four-momentum data [39, 51, 58, 73].

In the present analysis, we focus on classifying fat jets resulting from hadronically decaying boosted top quarks from light quarks and gluon jets (here onwards, QCD jets). Theoretically, the top quark is especially interesting because of its high Yukawa coupling. The large top Yukawa coupling plays not only a crucial role in the computation of electroweak precision observables [93] and determining the vacuum stability [94] of the SM, but also significantly influences the masses and interactions of several BSM resonances, many of which have enhanced couplings with the top quark, resulting in a top quark rich final state at the LHC. As the heaviest SM particle, the top quark decays into a $b$ quark and a $W^{\pm}$-boson. The subsequent $W^{\pm}$-boson decays can yield either a 3-quark final state or a combination of a $b$-quark and an SM-charged lepton, accompanied by missing transverse energy. Conventional searches at the LHC primarily rely on the leptonic decays of top quarks to suppress the huge SM QCD background. Although leptonic decays reduce the SM background contributions, the suppressed leptonic branching ratios of the top quark lead to reduced signal strength. Additionally, missing transverse energy from the elusive neutrinos in the final state complicates the reconstruction of the top quark's 4-momentum and the decay cascade of BSM resonances which lead to the top-rich final states. While the hadronic decays of the top quark simultaneously solve these two issues, the hadronic decay of the top quark into three resolved jets suffers from a huge QCD background. Effectively distinguishing these large-$R$ jets arising from boosted top quark decays from QCD jets is key to suppress QCD background for BSM scenarios featuring top

---

[2]Jet substructure observables are not only valuable for tagging boosted SM heavy particles like top quarks, Z/W-bosons, and Higgs bosons, but their significance in distinguishing between quark and gluon jets has also been demonstrated recently in the literature [12–18].

[3]While representation of a jet as a gray-scale (single layer) or color (multi-layer) image are used to train convolutional neural networks (CNNs) [40,41,44,52,54,55,65–71], Fisher discriminant analysis [56], locally connected networks [53], and Multi-Layer Perceptrons (MLPs) [50], the Graph Neural Networks GNNs [46,57,76–86] are trained on the particle cloud (graph) representation of jets. Similarly, jet-based tree-structured data [45,61,87,88] can also be used in Recursive Neural Networks (RvNNs) and GNNs.

quark-rich final states at the LHC. In this work, we have focused on three different machine-learning algorithms to distinguish top jets from QCD jets:

- A Boosted Decision Tree that uses high-level features for training.

- A miniature version of the ResNet [95] that uses image representation of jets as input.

- LorentzNet [76], a symmetry-preserving GNN founded upon the concept of Lorenz equivariance.

We have also considered composite classifiers by stacking a BDT-based classifier on top of the ResNet and LorentzNet to leverage the high-level features in BDT and low-level inputs in CNN/GNN in a single tagger.

The calorimeters at the LHC have a fixed granularity, and as the transverse momentum of top jets increases, the energy deposition by jet constituents becomes more compact. This compactness results in reduced resolution in variables constructed using calorimeter towers. To address this issue, researchers have turned to the finer spatial granularity of inner detectors, leveraging tracking information to improve their analyses. Moreover, the principles of Quantum Chromodynamics (QCD) and various experimental findings suggest that jets initiated by light quarks or gluons exhibit distinct differences in the distribution and composition of charged and neutral hadrons during their hadronization process.[4] Given this context, it is essential to investigate the impact of tracking information on top-tagging algorithms, as it can significantly enhance their performance.

This work will study the critical importance of combining information from calorimeter towers and the tracker detector. This combined information provides insights into the composition and distribution of charged and neutral hadrons within a jet, ultimately playing a crucial role in determining the performance of the classifiers. Additionally, we will study the effect of different Monte-Carlo generators on the performance of top tagging algorithms. We will also study the dependence of the classifier's performance on the transverse momentum of the fat jets.

The rest of the paper is organized as follows. In Section 2, we discuss the dataset used to train the classifiers. Section 3 discusses the various model architectures used for our analysis. In Section 4 and 4.5, we discuss the effect of tracking information and truth-level identification efficiency on classifier performance. In Section 4.6, we discuss the variation of classifier efficiency with the transverse momentum of the fat jets. Finally, Section 5 summarises our observations.

## 2 Dataset

A significant portion of our analysis focuses on establishing the importance of incorporating the information from tracker and calorimeter towers into the datasets[5] used for training and testing the Machine Learning (ML) based classifiers designed to identify hadronically decaying boosted top jets over the QCD light quark and gluon jets. To fulfill this objective, we have trained our classifiers on datasets generated following two different approaches. One dataset (denoted as $DATA_{calo}$ in the rest of the manuscript) only incorporates the information stored as energy deposits in the hadronic and electromagnetic calorimeter towers. The second dataset denoted as $DATA_{trck}$ extends the previous one by incorporating the information regarding the

---

[4]Exploiting the characteristics of light quark and gluon hadronization, several classifiers [96–98] based on tracking information have been developed for the classification of quark vs. gluon jets.

[5]By dataset, we imply the signal (hadronically decaying boosted top quark) and background (QCD generated quark and gluon) fat jets used for our analysis.

electric charge of the charged constituents from the trackers. In the former case, we have generated the datasets following the prescription of Ref. [99]. The authors of Ref. [99] have studied various top taggers and assessed their performance based on a dataset that only contains information on the jet constituents coming from the calorimeter energy deposits. Though that dataset[6] serves its purpose of comparing the performance of various top taggers, as we will demonstrate in the subsequent sections, the same dataset is inadequate in providing a given ML algorithm's optimal performance. To make our case, we have compared the performance of LorentzNet, a symmetry-preserving Graph Neural Network (GNN) for top tagging [76], on datasets generated using these two approaches. We have also performed similar exercises for the Boosted Decision Tree (BDT) and Convolutional Neural Network (CNN) based top taggers.

Another objective of this work is to study the performance of top taggers for different transverse momentum ranges of the hadronically decaying top quark. For this section of the analysis, we have divided the $p_T$ range between 300 to 1500 GeV into six bins of size 200 GeV each. We then generate large-$R$ jets resulting from the hadronically decaying boosted top quarks (the signal jets) and QCD production of light quarks and gluons (the background jets) in these bins and train and test our classifiers for each $p_T$ bin.

All large-$R$[7] signal and background fat jets are generated in MG5_AMC@NLO [106] with the NNPDF21LO [107] PDF. The hadronically decaying boosted top samples are generated from the SM process $pp \rightarrow t\bar{t}$. Similarly, for the background fat jets (QCD production of quark and gluon), we have used the process $pp \rightarrow jj$ (where $j$ includes $u, c, d, s, g$ and their anti-particles). Subsequent decay of the top quarks and showering and hadronization of the light quarks and gluons are simulated in Pythia8 [108]. To simplify the analysis, we have not included the effect of Multi Parton Interaction (MPI) and PileUp. Finally, we have used Delphes [109] with the default ATLAS card to include the detector efficiencies and resolutions. The fat jets are reconstructed in Fastjet [110] using the anti-kT algorithm. To reconstruct the sub-jets inside a fat jet, we use the jet trimming [111] algorithm with default parameters $R_{trim} = 0.2$ and $p_{T,trim} = 0.05$, which gives us subjets with $R = 0.2$. For each $p_T$ range defined in the previous paragraph, we have generated one million top quarks and one million QCD jets for our final analysis. For training purposes, we selected 600k fat jets from each category while reserving 200k from each class to validate and test the classifiers. For training and testing the composite classifiers (see section 3), we have generated additional 400k fat jets from each category for training and 200k from each category for testing.

Before proceeding to the next section, we mention the convention followed in our analysis to construct the constituents of a fat jet, namely the tracks and calorimeter towers. Throughout our analysis, we adopt two different conventions; in one, we use the TrackMerger/tracks and Calorimeter/towers classes of Delphes to construct the tracks and towers. We refer to them as tracks and towers in the subsequent sections. In the second convention, we use the HCal/eflowTracks to construct the tracks while we combine the ECal/eflowPhotons and HCal/eflowNeutralHadrons classes to construct the calorimeter towers. These are referred to as Etracks and Etowers in the subsequent discussion. The only difference between the two approaches is that, in the latter case, Delphes performs a matching between the track and calorimeter energy deposits to filter out the calorimeter towers originating from the charged particles and classify them as tracks.

---

[6]The same dataset have been used in several subsequent analyses [57,78,100–105] for accessing the performance of their proposed classifiers

[7]The reconstruction radius (radius of the cone used to define the fat jets in FastJet) plays a crucial role in determining the identification efficiencies of fat jets resulting from boosted top quarks. We will discuss this issue in the next section.

## 2.1 Truth-level tagging (TLT)

The quality of the training dataset has a big impact on how well a classifier performs. In our specific example, the classifier's performance is significantly influenced by the purity of the signal (hadronically decaying boosted top quarks) datasets.[8] Improved classifier efficiency results from using more pure signal data in training. Therefore, to prepare the signal samples for training and analyzing the performance of any classifier, we need a method to select only those fat jets that are properly reconstructed. We achieve this objective by matching the fat jets and the constituent sub-jets with their partonic counterparts. This process is referred to as truth-level tagging (TLT), and we name the efficiency of a hadronically decaying boosted top quark to be associated with a properly reconstructed fat jet as the truth-level identification efficiency ($\epsilon_S^{truth}$). To ensure a valid comparison with the performance of the available classifiers in the existing literature, we adopt the following simple truth-level tagging criteria, introduced in Ref. [99] and subsequently employed in several other references [57, 78, 100–105]: A fat jet to be tagged as a top fat jet at truth level, we require that both the partonic top and its three daughter quarks lie within the cone of that fat jet. No truth-level tagging criteria are applied to the QCD fat jets.

The truth-level identification efficiency depends on two factors: the reconstruction radius ($R$) of the fat jet and the transverse momentum of the hadronically decaying top quark. If the transverse momentum of the top quark is not large enough, the decay products of the top quark will not be collimated enough, and we will require a large radius fat jet to capture all the hadrons arising from the hadronization of the three light quarks resulting from top decay. At the same time, if we have a top quark with very high transverse momentum, all the hadronic constituents resulting from the high-$p_T$ top quark will reside inside a small cone. In this case, if we choose a very large radius of reconstruction, the fat jet will pick contributions from the background radiation, which will directly affect the resolution of various features of the fat jet and hence, the performance of the classifiers. The way out is to use a jet tagging algorithm with a variable radius of reconstruction [112, 113], which is beyond the scope of our analysis. Instead, we work with different reconstruction radii for fat jets in the six transverse momentum regimes mentioned in the previous section. In Figure 1, we present the variation of truth-level identification efficiency ($\epsilon_S^{truth}$) with the radius of fat jets for the six $p_T$ bins. For our final analysis, we choose the reconstruction radius for the top fat jets in the different $p_T$ bins such that we can achieve a notable $\epsilon_S^{truth}$ without being concerned about the distortion of crucial jet characteristics caused by background radiation. Ergo, we reconstruct fat jets in [300, 500] GeV and [500, 700] GeV $p_T$ bins with a $R = 1.2$. While for the remaining four $p_T$ bins, we fix the value of $R$ at 0.8.

## 2.2 Extracting the data

The classifiers addressed in the remaining manuscript can be categorized into three primary groups: Boosted Decision Tree (BDT-classifiers), Convolutional Neural Network (CNN-classifiers), and Graph Neural Network (GNN-classifiers). The nature and structure of training and testing datasets for these three classifier groups differ significantly. While the BDT classifiers use high-level variables/features (HLF) (invariant mass of the fat jet, N-subjettiness, etc.) as input, GNN or CNN classifiers, on the other hand, use low-level features (LLF) such as the four-momentum of the constituents or jet image constructed from the transverse momentum of the jet constituents. We will discuss the nature and structure of the datasets used for training and testing these three groups of classifiers in the following.

---

[8]One or more constituents may reside outside the jet reconstruction cone, rendering the signal sample impure. For instance, in the case of a fat top jet, the final fat jet is fundamentally a W-jet rather than a top-jet if the b-quark sits outside the reconstruction cone.

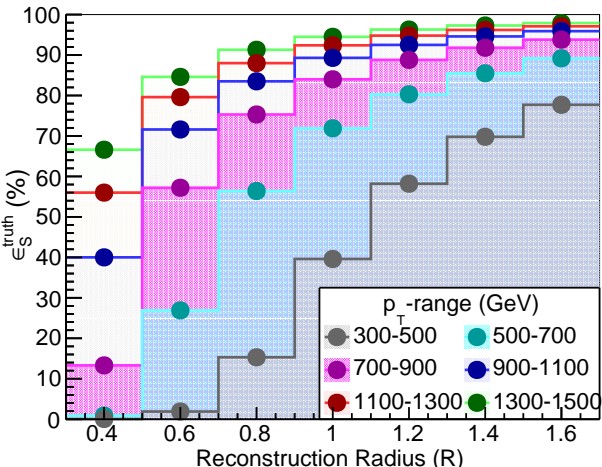

Figure 1: The variation of truth-level identification efficiency with the jet radius in different transverse momentum ranges.

### 2.2.1 BDT

The BDT uses high-level variables/features (HLFs)[9] for classification. For each signal and background fat jet, we use the information of the constituent tracks and calorimeter towers to construct the desired high-level features.[10] In this work, we studied two different BDT-based classifiers. The first BDT classifier, referred to as the tower-based BDT classifier or $BDT_{calo}$, utilizes the five most commonly used high-level features (HLFs) for top tagging: the invariant mass, three ratios of N-subjettiness variables, and the b-tagging information. In the case of the second BDT classifier, referred to as the track-based BDT classifier or $BDT_{trck}$, supplementary HLFs are designed using information from the tracker detector. The goal of the tracker detector at the LHC is to trace the paths of charged hadrons, thereby offering insights into the electrically charged constituents of the jet. The HLFs used for $BDT_{calo}$ and $BDT_{trck}$ classifiers are discussed in the following:

As discussed above, the $BDT_{calo}$ uses five HLFs,

- **The invariant mass** of the fat jets :

$$M = \sqrt{\sum_i (E_i)^2 - \sum_i (p_i)^2},$$

(1)

where the sum runs over all constituents of the fat jet.

- **The N-subjettiness variable $\tau_N$** :

$$\tau_N = \frac{1}{\sum_k p_{Tk} R_0^\beta} \sum_k p_{Tk} min(\Delta R_{1,k}^\beta .. \Delta R_{N,k}^\beta),$$

(2)

where the sum runs over the constituents with transverse momentum $p_{T,k}$, $R_0$ is the radius parameter used in the clustering algorithm, $\beta = 1$ is the thrust parameter, and

---

[9]These are functions of low-level variables like the four-momentum and position in the $\eta - \phi$ plane of the jet constituents.

[10]For this section of our analysis, we have used the track and tower class of Delphes to construct the fat jet constituents.

$\Delta R_{i,k}$ characterize the separation between the constituent $k$ and the candidate sub-jet $i$. In our analyses, we use three ratios of the N-subjettiness variables $\tau_{43}$, $\tau_{32}$, and $\tau_{21}$, where $\tau_{mn} = \tau_m/\tau_n$.

- **b-tag**: We consider a fat jet b-tagged when there is at least one b-tagged $\Delta R = 0.4$ sub-jets inside the cone of the fat jet, i.e., $\Delta R(J, j_b) < R_0$.

In the $BDT_{trck}$ classifier, we extend the above list by including 21 HLFs. Most of these HLFs are discussed in [18, 98, 114–117], we summarise them here for completeness:

- $N_{trk}$: It characterizes the number of tracks inside a jet.

- $w_{trk}$: The $p_T$ weighted width of the tracks:

$$w_{trk} = \frac{\sum_{trk \in J} p_{T,trk} \Delta R_{trk,J}}{\sum_{trk \in J} p_{T,trk}} . \tag{3}$$

- $w_{calo}$: the $E_T$ weighted width, defined as:

$$w_{calo} = \frac{\sum_{i \in J} p_{T,i} \Delta R_{i,J}}{\sum_{i \in J} p_{T,i}} , \tag{4}$$

where the sum runs over the jet constituents with transverse momentum $p_{T,i}$.

- $E_{frac}$: the ratio of the energy of the hardest constituent to the jet's energy:

$$E_{frac} = \frac{E_{hardest}}{E_J} . \tag{5}$$

- $C_\beta$: the two-point energy correlation function:

$$C_\beta = \frac{\sum_{i,j \in J} E_{T,i} E_{T,j} (\Delta R_{i,J})^\beta}{(\sum_{i \in J} E_{T,i})^2} . \tag{6}$$

For our analysis, we use a value of 0.2 for $\beta$.

- **The Jet Charge**: the $p_T$ weighted sum of the charge of the constituent tracks:

$$Q_k = \frac{\sum_i q_i (p_{Ti})^k}{\sum_i p_{T,i}} , \tag{7}$$

where k, the regularisation exponent, has a value of 1 for our analysis.

- $\Delta R_{sub}$: The $\Delta R$ separation between the sub-jets inside a fat jet. They constitute a set of three variables $\Delta R_{1,2}$, $\Delta R_{2,3}$, and $\Delta R_{3,1}$. The numbers in the subscript denote the $p_T$-ordered sub-jets.

Note that the variables ($N_{trk}$, $w_{trk}$, $w_{calo}$, $E_{frac}$, $C_\beta$, and $Q_k$) are defined for each sub-jet inside a fat jet. For our analysis, we store the information of the first three highest $p_T$ sub-jets. Therefore, they constitute a set of 18 variables for each fat jet. The absent variables are zero-padded for a fat jet with less than three sub-jets.

In summary, the $BDT_{calo}$ classifier uses a set of five variables, and the $BDT_{trck}$ classifier uses a list of 26 variables of which five are the ones used in $BDT_{calo}$, three are the $\Delta R$ separation between the three highest $p_T$ subjets and six features ($N_{trk}$, $w_{trk}$, $w_{calo}$, $E_{frac}$, $C_\beta$, and $Q_k$) for each of these three subjets.

### 2.2.2 CNN

The Convolutional Neural Network (CNN) uses grid-shaped data or images for classification tasks. The units in an image are referred to as pixels, and each pixel is associated with the pixel intensity. For our analysis, we have used the transverse energy[11] of the tracks and towers as pixel intensities. As mentioned in Section 2, we use two different datasets to demonstrate the importance of tracking information in enhancing the performance of the classifiers.

The first dataset only uses the information of the calorimeter energy deposits[12] to construct the images. Therefore, these images have only one layer and are of dimension 64 × 64. The process of constructing these images is slightly different than the conventional methods. We demonstrate this with a simple example. Suppose we have a fat jat with R=0.8. If we convert it into an image with dimension 64 × 64, we end up with pixels of dimension 0.025×0.025 — significantly smaller than the HCal resolution. To circumvent this, we first divide the jet into pixels of size 0.1×0.1, commensurate with the HCal resolution. This will result in an image with dimension 16×16. To get the final image with dimension 64×64, we further divide each pixel of intensity $E_{T,i}$ into a 4×4 grid where each final pixel caries an intensity $E_{T,i}/16$. In the subsequent discussions, we refer to the CNN trained on this dataset as $CNN_{calo}$.

In the second dataset, we use the information of both tracks and calorimeter towers to construct a two-layer image of dimension 2 × 64 × 64. Here, we make use of the Etrack and Etower classes of delphes. We adhere to the above image generation procedure for the layer constructed from the Etower class. The situation is, however, different for the layer constructed from the Etrack class. At LHC, the position of the tracks in the $\eta-\phi$ plane can be measured with high accuracy. This allows us to split the jet directly into a 64 × 64 image. In the subsequent discussions, we named the CNN trained using the second dataset $CNN_{trck}$.

To boost our taggers' performance, we process each image using a similar method as described in [40,118]. The pre-processing steps make use of the sub-jets inside a fat jet. In Figure 2, we present the evolution of top and QCD images[13] through subsequent preprocessing stages. For better comparison, we present the top and QCD images side-by-side. First, we centralize the images such that the sub-jet with the highest $E_T$ shifts to the origin of the $\eta-\phi$ coordinate system (see the first row of Figure 2). We see a widespread distribution of constituents in the top image. The energy in the QCD jets is distributed near the center, demonstrating its origin from a single parton. Next, we rotate the image so that the next-to-highest $E_T$ sub-jet lies below the first sub-jet. In the absence of a second sub-jet, we rotate the image around the jet-energy centroid until the image's principal axis [56,119] becomes vertical. We present the resulting average images in the second row of Figure 2. We see the clear appearance of a second hard structure for the top jet and a diffusive radiation pattern for the QCD jets. Then, we reflect the image such that the sum of pixel intensities on the right-hand side of the image

---

[11]The transverse energy is defined as $\frac{E}{\cosh \eta}$.

[12]Here, we utilize the tower class of delphes to reconstruct the constituent information of the fat jets.

[13]The images demonstrated here result from averaging over 10000 individual images. This averaging makes the structures in the image more visible.

is higher than that on the left-hand side (third row of Figure 2). Finally, we normalize the image by dividing each pixel intensity by the sum of the intensities of all pixels.

### 2.2.3 GNN

Like CNN, we construct two datasets to study the performance of GNN. The first dataset uses the tower class of Delphes to construct the jet constituents, while the second uses the Etrack and Etower classes. In both these cases, we store the four-momentum of the first 200 highest $p_T$ constituents and their charge for each fat jet. If the fat jet has less than 200 constituents, we fill the remaining entries with zero. For the Etowers and towers, we set the charge to be zero, while for the Etracks, the charge can take value $\pm 1$. There is one important point to note: at the LHC, the mass of the tracks is measured from the curvature in the magnetic field and the momentum of the tracks. Later they match this mass with the mass of a physical particle by following a matching scheme. Nevertheless, we have refrained from incorporating details regarding the particle identity of the charged track. Instead, our approach solely relies on the electric charge information of the constituents. This decision not only diminishes the classifier's sensitivity to the specifics of the hadronization model but also minimizes uncertainties stemming from the tagging or mis-tagging efficiencies of charged hadron identities. In the subsequent discussions, we follow a simplified approach and make the Etracks massless by hand to mask the identity of the charged hadron to the classifiers, i.e., we only use the information of the three momenta of the Etracks and set the energy as the magnitude of three momenta.

## 3 Models

In this section, we will discuss the architecture of the Machine Learning (ML) classifiers used in our analysis. We have organized our discussion in order of the complexity of the ML classifiers. First, we discuss a simple cut-based classifier, the Boosted Decision Tree. Next, we discuss the architecture of a CNN classifier that works with image-shaped data. Finally, we will demonstrate the architecture of a Graph Neural Network (GNN) where the input is graph-structured data.

### 3.1 BDT

The $BDT_{calo}$ (see 2.2.1) classifier uses a small set of HLFs, emphasizing the importance of invariant mass, N-subjettiness variables, and b-tagging information in discriminating signal top fat jets from QCD light quark and gluon background jets. The classifier $BDT_{trck}$ focuses more on the HLFs resulting from identifying the jet's charged constituents from the tracker detector. In addition to the above variables, $BDT_{trck}$ includes several other track-based features that characterize the composition of charged and neutral hadrons inside the sub-jets and the fat jet. For a consistent performance comparison, both BDTs have the same hyper-parameters and are trained using the TMVA 4.3 toolkit [120] integrated into ROOT 6.24 [121] analysis framework. Table 1 summarizes these hyper-parameters.

### 3.2 CNN

The CNN model used in our analysis is a miniature version of the original ResNet model [95]. The ResNet architecture was originally designed to solve the vanishing gradient problem in very deep Neural Networks. ResNet uses the principle of residual connections that allows it to maintain a stable gradient propagation throughout the network. This residual/skipped connection passes the input of the ResNet block directly to the output along with the learned features. Mathematically,

$$x_{i+1} = x_i + F(x_i), \tag{8}$$

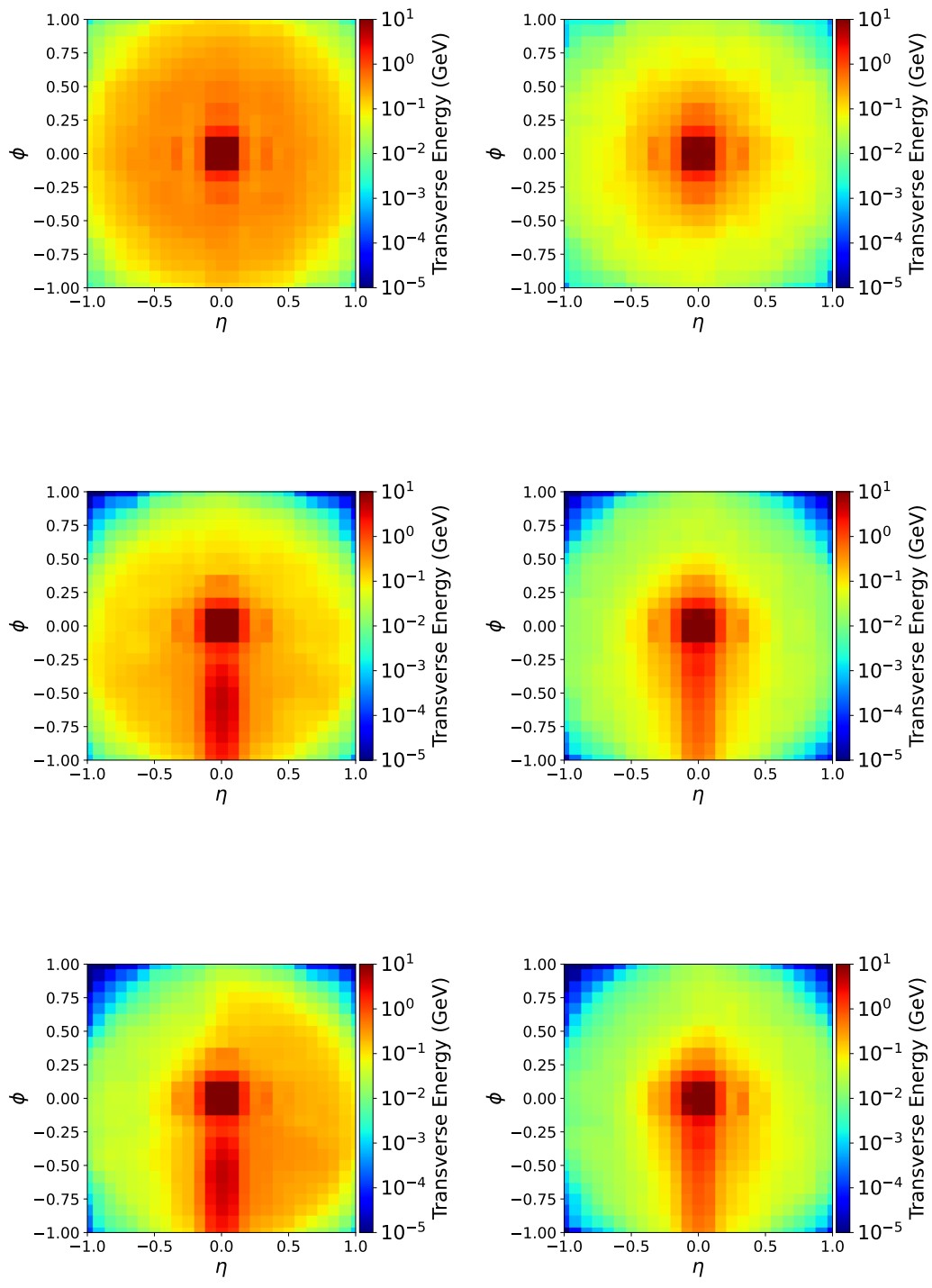

Figure 2: Different image preprocessing stages of the top image (left) and the corresponding QCD image (right). The first row represents the images after the translation preprocessing step, the second row presents the images after rotation, and the third row shows the images after reflection. More on these preprocessing steps is provided in the text.

Table 1: Summary of optimised BDT hyperparameters.

| BDT hyperparameter | Optimised choice |
| --- | --- |
| NTrees | 1000 |
| MinNodeSize | 5% |
| MaxDepth | 4 |
| BoostType | AdaBoost |
| AdaBoostBeta | 0.1 |
| UseBaggedBoost | True |
| BaggedSampleFraction | 0.5 |
| SeparationType | GiniIndex |
| nCuts | 40 |

where $x_i$ represents the input to the ResNet Block, $x_{i+1}$ is the output, and $F(x_i)$ represents the residual function, a collection of non-linear operations.[14] The architecture of the ResNet block and the full CNN model is presented in Appendix A.

The ResNet model is trained with PyTorch on a single Nvidia Tesla K80 GPU. The model is trained for 35 epochs with a batch size of 32. We use the ADAMW [122] optimizer with a weight decay of $10^{-2}$ and an initial learning rate of $10^{-3}$ to minimize the Cross-Entropy loss function. We reduce the learning rate by half for the first five epochs. After that, the learning rate is reduced at a rate of 10%, and for the last five epochs, we reduce the learning rate by 90 % per epoch. We check the model's performance after every epoch on the validation dataset, and the model with the best validation accuracy is used for the final test.

### 3.3 GNN

We use the LorentzNet [76], a symmetry-preserving deep Neural Network, for the GNN part of our analysis. LorentzNet utilizes the Lorentz group equivariance principle [78] to construct the Neural Network's layers. This means under Lorentz transformation, the output of the neural network follows the transformation of the input, i.e.,

$$x \rightarrow F(x), \quad and \quad \Lambda(x) \rightarrow F(\Lambda(x)) \implies F(\Lambda(x)) = \Lambda F(x). \tag{9}$$

Here, $x$ is the input to the neural network layer, $F(x)$ is the output, and $\Lambda$ represents the Lorentz transformation.

The graph neural network operates on graph-structured data [123, 124]. A graph is a collection of nodes and edges, i.e., G(V, E), where $V = x \oplus h$ denotes the nodes, and E denotes the edges between the nodes. Each node is characterized by a node coordinate x, which in our case is the four-momentum of the jet constituents, and a node attribute/embedding h, which for our analysis is the charge of the constituents. LorentzNet does not assume any prior knowledge regarding the relationship between the nodes. In other words, it uses fully connected graphs. For a detailed discussion on the model, its implementation, the optimizer used, and the learning rate scheduler, See [76].

We implemented the LorentzNet with PyTorch and trained it on a cluster with four Nvidia Tesla K80 GPUs. We pass the data in batches of size 16 on each GPU. The model is trained for a total of 35 epochs. At the end of each epoch, we test the model performance with the validation dataset, and the one with the best validation accuracy is saved for testing.

---

[14]The Convolution and Normalisation operations are few examples.

## 3.4 Composite models

So far, we have discussed six different classifiers denoted as simple in the later part of the manuscript. A simple $BDT_{calo}$ that only uses the features extracted from the calorimeter energy deposits of a fat jet without considering the tracking information. Next, we have an extended version of the simple $BDT_{calo}$, the simple $BDT_{trck}$ classifier, which extends the previous dataset by including complementary information from the tracking detectors. Then, we discussed the one-dimensional $CNN_{calo}$ and $GNN_{calo}$ classifiers, which only use the information of the calorimeter towers inside a fat jet. The $CNN_{trck}$, on the other hand, uses 2-dimensional images where the second layer comprises the tracks that constitute the fat jet. Similarly, we have $GNN_{trck}$, which uses the charged hadrons and neutral hadrons four-momentum from tracks and towers to construct particle clouds/graphs.

We expect that during training, the CNN/GNN can extract important characteristics of the fat jets from these low-level features that can discriminate between the signal and background jets. However, some information about the high-level features is lost during the data pre-processing, which can be extremely valuable for the classification task. For example, as demonstrated in [55], the rotation and normalization preprocessing steps in generating the images for CNN smear the information of the invariant mass of a fat jet. Similarly, the b-tagging information of a fat jet is not included in the CNN and GNN classifiers but can be useful for the classification task. The BDTs also have one obvious disadvantage. They rely on the user-supplied HLFs rather than extracting features directly from data. This limits their ability to automatically learn the complex features present in the data.

From the above discussion, it is clear that the simultaneous use of both LLFs and HLFs can help explore complementary directions in the feature space and improve the performance of the classifiers. One naive way of incorporating both HLFs and LLFs in an analysis is to stack classifiers that use these features on top of one another. We refer to them as composite classifiers. The idea is first to use a classifier (a CNN/GNN) that uses LLFs to extract a preliminary classification score and later use this score as an HLF in a second classifier (a BDT) along with other HLFs. In this work, we have studied the performance of eight such composite Models; $CNN_{calo}+BDT_{calo}$ ($C_{calo}B_{calo}$), $CNN_{calo}+BDT_{trck}$ ($C_{calo}B_{trck}$), $CNN_{trck}+BDT_{calo}$ ($C_{trck}B_{calo}$), $CNN_{trck}+BDT_{trck}$ ($C_{trck}B_{trck}$), $GNN_{calo}+BDT_{calo}$ ($G_{calo}B_{calo}$), $GNN_{calo}+BDT_{trck}$ ($G_{calo}B_{trck}$), $GNN_{trck}+BDT_{calo}$ ($G_{trck}B_{calo}$), and $GNN_{trck}+BDT_{trck}$ ($G_{trck}B_{trck}$). In the next section, we will demonstrate the performance of all these models in discriminating top jets from QCD jets.

## 4 Classifier performance

In this section, we will discuss the performance of the different classifiers for top tagging. For a consistent comparison with the results of [57, 76, 78, 99–105], we generate top and QCD samples in the 550 $GeV < p_T < 650\ GeV$ range and reconstruct the fat jets with $R = 0.8$. The generation process is the same as discussed in Section 2. At the same time, to check the dependency of the classifier performance on the showering and hadronization models of the Monte Carlo event generator, we have generated a second QCD sample[15] using Herwig [125,126]. We train the classifiers using the Pythia-generated dataset and save the model that performs best on the validation set for further analysis. We perform two final tests, one using the Pythia-generated signal and background sample and the other where the signal jets are generated using Pythia while background jets are generated using Herwig.

---

[15]The reason for this choice lies in the truth-level identification efficiency. The parton-level information in a Herwig-generated dataset differs from that in a Pythia-generated sample. This results in different TLIEs. Subsequently, the resulting top samples are inadequate for comparing the classifiers' performance. However, since we do not perform any truth-level identification for the QCD jets, they can be used for the task.

To facilitate a better understanding of the discussion, below, we present the definitions of the different tagging efficiency frequently used in our paper:

- $\epsilon_S^c$ refers to the classifier efficiency, i.e., the faction of truth-tagged top jets identified by the tagger as top jets.

- $1/\epsilon_B^c$ refers to the background rejection associated with the classifier efficiency. Here, $\epsilon_B^c$ represents the fraction of QCD jets that get mis-tagged as top jets by the tagger. Note that we have not used any truth-level tagging criteria for the QCD jets in our analysis.

- $\epsilon_S^{tag}$ refers to the top-tagging efficiency, i.e., the fraction of top jets (generated without any truth level tagging criteria) identified by the classifier as top jets.

- $1/\epsilon_B^{tag}$ refers to the background rejection associated with the top-tagging efficiency.

In Figure 3, we present the performance of the classifiers in the form of their Receiver Operator Characteristic (ROC) curves. The solid lines represent the ROC curves for the dataset where both signal and background samples are generated using Pythia. On the other hand, the dotted curves characterize the sample where the background jets are generated in Herwig. In Table 2, we present the background rejection of all the classifiers corresponding to 70 and 50 % classifier efficiency (second and third column) as well as top-tagging efficiency (fourth and fifth column). The background rejection within the parentheses in Table 2 represents the results obtained from the dataset simulated in Herwig. This section will address the results in the second and third columns. For a detailed discussion of the results in the fourth and fifth columns, see Section 4.5. In the following discussion, we will use the background rejection at 50% classifier efficiency ($\epsilon_S^c$) as a metric to compare the performance of different simple and composite classifiers introduced here and also with existing top taggers in the literature.

Before delving extensively into the discourse of comparing the performance of various simple and composite classifiers, validating our approach (event simulation, sample preparation, etc.) by comparing our results with existing literature is crucial. We have provided a brief discussion on the validity of our analysis in Appendix B.

## 4.1 The performance of tower-based simple classifiers

The top left plot of Fig. 3 shows the ROC curves for $BDT_{calo}$, $CNN_{calo}$, and $GNN_{calo}$ classifiers in green, red, and blue, respectively. From the ROC curves, we can make the following comments:

- Comparing the performance of $GNN_{calo}$ with $CNN_{calo}$ and all LLF-based classifiers in Ref. [99] we can safely conclude that GNN classifiers like the LorentzNet [76] or $GNN_{calo}$ perform better than other LLF-based classifiers.

- Interestingly, the simple HLF-based $BDT_{calo}$ classifier has a comparable performance with the $GNN_{calo}$ and it performs better than $CNN_{calo}$ and all the CNN and GNN-based classifiers presented in Refs. [76, 99].

The poor performance of $CNN_{calo}$ is because the jet image preprocessing steps described in Section 2 dilute the jet mass information, an extremely important discriminant for classifying top jets over QCD jets. The $GNN_{calo}$, despite using complete 4-momentum information of the jet constituents, is trained using the calorimeter tower data. In contrast, the HLFs for training the $BDT_{calo}$ are derived from the fat jets constructed with Etracks and Etowers (see section 2). The superior energy/momentum resolution of tracks results in better performance for the $BDT_{calo}$ compared to $GNN_{calo}$.

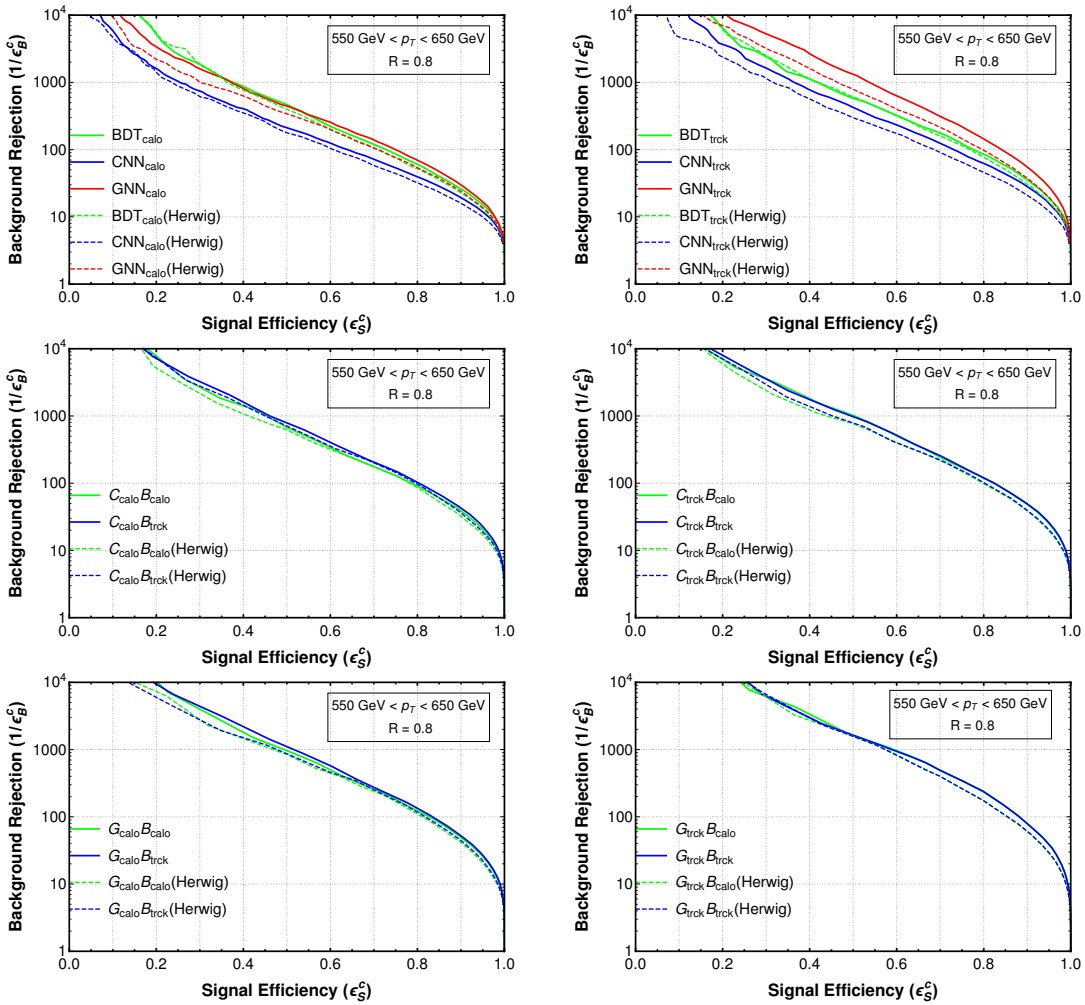

Figure 3: Thr ROC curves for the different classifiers for top and QCD samples in the $p_T$ range 550-650 GeV. The solid lines correspond to signal and background samples generated using Pythia, while the dotted line corresponds to the case where the background sample is generated using Herwig

## 4.2 Comparision of track and tower-based simple classifiers

Figure 3 (top right panel) shows the performance of the BDT, CNN, and GNN trained and tested using DATA$_{trck}$[16] namely the $BDT_{trck}$, $CNN_{trck}$, and $GNN_{trck}$ classifiers for both Pythia and Herwig-generated datasets. We see a significant performance boost—over 100%—for CNN and GNN-based classifiers trained and tested with DATA$_{trck}$ compared to tower-based classifiers (top left panel of figure 3). There are two main reasons for this improvement:

1. The use of high-quality datasets (DATA$_{trck}$) resulting from the superior resolution of the LHC tracker detector.

2. Incorporating data on both neutral and charged hadron compositions of fat jets into the training and testing datasets that enables the classifiers to differentiate substructures originating from the hadronization of a partonic gluon (in QCD jets) and a light quark (in the top decay product) more accurately. It has already been known, both from theoretical

---

[16]DATA$_{trck}$ was defined in section 2.

Table 2: Background rejection at 50 and 70 % background efficiencies. The terms in the bracket represent the results for the Herwig-generated dataset. The second and third columns correspond to a dataset where the top samples satisfy the truth-level tagging criteria. For the fourth and fifth columns, no such tagging criteria are imposed.

| Classifier | $1/\epsilon_B^c(\epsilon_S^c = 0.7)$ | $1/\epsilon_B^c(\epsilon_S^c = 0.5)$ | $1/\epsilon_B^{tag}(\epsilon_S^{tag} = 0.7)$ | $1/\epsilon_B^{tag}(\epsilon_S^{tag} = 0.5)$ |
|---|---|---|---|---|
| $BDT_{calo}$ | 119(105) | 467(398) | 22 | 125 |
| $CNN_{calo}$ | 70(57) | 211(178) | 17 | 76 |
| $GNN_{calo}$ | 139(106) | 444(341) | 24 | 139 |
| $BDT_{trck}$ | 175(159) | 579(610) | 33 | 180 |
| $CNN_{trck}$ | 124(90) | 423(299) | 25 | 120 |
| $GNN_{trck}$ | 311(214) | 1322(789) | 42 | 274 |
| $C_{calo}B_{calo}$ | 176(175) | 682(619) | 31 | 179 |
| $C_{calo}B_{trck}$ | 208(204) | 811(737) | 35 | 222 |
| $C_{trck}B_{calo}$ | 249(218) | 1023(768) | 43 | 253 |
| $C_{trck}B_{trck}$ | 257(221) | 995(799) | 46 | 249 |
| $G_{calo}B_{calo}$ | 260(241) | 969(842) | 43 | 261 |
| $G_{calo}B_{trck}$ | 278(256) | 1141(894) | 52 | 281 |
| $G_{trck}B_{calo}$ | 489(397) | 1641(1604) | 65 | 468 |
| $G_{trck}B_{trck}$ | 493(399) | 1736(1666) | 68 | 500 |

principles[17] and a large collection of experimental measurements [128–131], that jets initiated by gluons differ significantly from those initiated by light-flavor quarks. For example, gluon jets have higher charged particle multiplicity, a softer fragmentation function, are less collimated than quark jets, etc. Including the tracker data allows the classifier to leverage these differences and enhance its performance.

An evident drawback of developing a tool reliant on the hadronization of light quarks and gluons is the inherent discrepancies in the modeling of quark and gluon jets in Monte Carlo simulations. However, event generators like Pythia and Herwig incorporate sophisticated experimentally fine-tuned models for hadronization developed through decades of experimental studies and perturbative QCD calculations. Despite minor discrepancies between these event generators (as well as between the event generators and experimental data), giving rise to the systematic uncertainty, hadronization models used in Pythia and Herwig serve as a solid foundation for building improved classifiers for top tagging.

Figure 3 (top panel) also shows around 25 % improvement in the performance of $BDT_{trck}$ compared to $BDT_{calo}$. This can be ascribed to the use of subjet-based features constructed from the track and calorimeter tower constituents of the fatjet. To illustrate this point, we present in Appendix B the ranking[18] and covariance matrix[19] of a few important variables used in training the BDT classifiers. Appendix B shows that along with the jet mass, the features of

---

[17]The fundamental principle underlying the differentiation between quark and gluon jets is rooted in the observation that gluon splitting is stronger than quark splitting, as dictated by perturbative QCD. This distinction becomes evident by directly comparing the splitting probabilities for gluons, such as $g \to gg$ and $g \to u\bar{q}$, with those for quarks, like $q \to qg$ [127]. Therefore, on average, gluon jets are broader and encompass a higher particle multiplicity than quark jets with similar $p_T$.

[18]The variable ranking demonstrates the importance of the variables for the classification.

[19]Two variables that are least correlated represent independent directions in the feature space and, when used simultaneously, can considerably improve the performance of a classifier.

the second sub-jet play a crucial role in the classification task. Figure 3 (top right panel) also shows a comparable performance between the $BDT_{trck}$ and $CNN_{trck}$ classifiers. As discussed in Section 3.4, the pre-processing steps in CNN smear the invariant mass distribution for the fat jets, which plays a key role in the discrimination of top from QCD jets. Therefore, performances of simple CNN classifiers ($CNN_{calo}$ and $CNN_{trck}$) can be improved significantly when used in association with BDT classifiers. Potential improvements in such composite classifiers will be explored in the next section.

## 4.3   The performance of composite classifiers

Figure 3 presents the ROC curves for the composite CNN ($C_{calo}B_{calo}$, $C_{calo}B_{trck}$, $C_{trck}B_{calo}$, and $C_{trck}B_{trck}$, middle panel) and GNN ($G_{calo}B_{calo}$, $G_{calo}B_{trck}$, $G_{trck}B_{calo}$ and $G_{trck}B_{trck}$, bottom panel) classifiers (see section 3.4 for definitions). Incorporating the lower-level information of CNN with the higher-level information of BDT improves CNN performance by over 100%. As discussed in the previous section, the information on the HLFs, like invariant mass and b-tag, are not present in the CNN score. Therefore, when used together in a composite classifier, they significantly enhance performance.

Combining the GNN score with HLFs from $BDT_{calo}$ improves $GNN_{calo}$ performance by about 100% in $G_{calo}B_{calo}$ and $GNN_{trck}$ performance by about 25% in $G_{trck}B_{calo}$ (see Table 2). The marginal enhancement in performance observed for $G_{trck}B_{calo}$ can be attributed to the fact that the training datasets for $GNN_{trck}$ already encompass comprehensive information about the constituent tracks and towers, making additional HLFs from $BDT_{calo}$ less impactful. To illustrate this, we present the ranking of HLFs used in $G_{trck}B_{calo}$ and $G_{calo}B_{calo}$ in Table 12 of Appendix D. In contrast to $G_{calo}B_{calo}$, where the highest-ranked variable is the invariant mass of the fat jets (and consequently, it is frequently employed for node splitting), the GNN score takes the top-ranking position in $G_{trck}B_{calo}$, demonstrating the importance of this variable.

While $C_{calo}B_{trck}$ shows around 20% performance boost over $C_{calo}B_{calo}$, attributed to the complementary nature of the track-based HLFs in $BDT_{trck}$ alongside the calorimeter tower-based LLFs used in $CNN_{calo}$, the performance of $C_{trck}B_{trck}$ is comparable to $C_{trck}B_{calo}$. This is again due to the training dataset of $CNN_{trck}$ that already includes all the relevant information from both tracks and towers for classification. We see almost similar behavior for the track and tower-based composite GNN classifiers.

We expect $GNN_{trck}$ with the full tracking information to be efficient enough to extract all relevant features of the fat-jet and hence to have a similar performance as $G_{trck}B_{calo}$ and $G_{trck}B_{trck}$. Surprisingly, composite GNN classifiers still significantly outperform ordinary track-based GNN classifiers. This behavior is due to masking the mass information of the fat-jet constituents arising from tracks (See Section 2.2.3). To validate this point, we retrain the $GNN_{trck}$ classifier with a dataset with unmasked tracking information and present our findings in Table 3. As per our expectation, the $GNN_{trck}$ classifier with full unmasked tracking information has a comparable performance with $G_{trck}B_{trck}$. So, is it really necessary to introduce a composite classifier when we can achieve the same performance by training the original classifier with the complete information? To investigate the matter further, we performed another study where instead of testing the classifiers on a pythia-generated dataset, we tested it on data where the Background sample is generated using Herwig. The motivation for this study is to check the effect of systematic uncertainties arising from using different Monte-Carlo event generators on the performance of the classifiers. As it is evident from the second row of Table 3, the Composite track-based GNN classifier has better control over these uncertainties than the ordinary GNN classifier trained with full unmasked tracking information. We present a detailed discussion in the next section.

Table 3: The first row presents the background rejection at 50% classifier efficiency for $GNN_{trck}$ classifier trained and tested using a dataset with the full unmasked tracking information and that of the $G_{trck}B_{trck}$ classifier trained and tested with the masked dataset. The second row represents similar results, but the testing is performed using a background sample generated using Herwig7.

| MC generator | $GNN_{trck}$ | $G_{trck}B_{trck}$ |
|---|---|---|
| Pythia8 | 1769 | 1736 |
| Herwig7 | 1025 | 1666 |

## 4.4 Systematic uncertainties of different simple and composite classifiers

To investigate the systematic uncertainties arising from the showering and hadronization models of event generators, we evaluated the performance of classifiers using datasets generated by Pythia and Herwig. The solid and dotted lines in each plot of Figure 3 represent the ROC curves corresponding to Pythia and Herwig, respectively, generated testing datasets for a specific classifier.

The performance of the BDT classifiers ($BDT_{calo}$ and $BDT_{trck}$), which rely on HLFs, remains largely unaffected by the choice of Monte-Carlo generators. These HLFs exhibit minimal sensitivity to jet modeling, leading to reduced systematic uncertainties for BDT classifiers. In contrast, due to the direct correlation between the LLFs of the fat jets and the jet modeling within the Monte-Carlo generators, CNN/GNN classifiers trained on Low-Level Features (LLFs) of jets show significant sensitivity to the showering and hadronization model used, resulting in large systematic uncertainties (as depicted in the top panel of Figure 3). For tower (track) based CNN/GNN classifiers, these uncertainties can reach up to 30% (40%). The higher sensitivity of track-based CNN/GNN classifiers to the jet modeling of the Monte-Carlo generator results from the fact that these classifiers are trained on $DATA_{trck}$ that encompass the finer details of the showering and hadronization processes. In contrast, for tower-based classifiers, the limited resolution of the calorimeter smooths out some of these dependencies.

Remarkably, composite classifiers not only enhance top-tagging performance but also show reduced systematic uncertainties compared to simple LLF-based classifiers. Note that in composite classifiers, scores from LLF-based CNN/GNN classifiers are treated as additional HLF in conjunction with other HLFs of the track and tower-based BDTs discussed in section 2.2.1. Interestingly, while the scores from LLF-based classifiers introduce higher systematic uncertainties, other HLFs of the BDT classifiers remain relatively insensitive to variations in Monte-Carlo generators.

The ranking of HLFs used in composite classifiers, as depicted in Tables 10, 11, 12 and 13 of Appendix D, illustrates that alongside the scores from LLF-based classifiers, the other HLFs also make substantial contributions to the classification task. For instance, for the composite classifiers like $C_{calo}B_{calo}$, $C_{trck}B_{calo}$, $C_{calo}B_{trck}$, $C_{trck}B_{trck}$, $G_{calo}B_{calo}$, and $G_{calo}B_{trck}$, jet mass holds the highest ranking among the HLFs. The utilization of the LLF-based score as a classifying feature is restricted to about 36 % (19 %) for $G_{trck}B_{calo}$ ($G_{trck}B_{trck}$), where score takes precedence as the highest ranking variable. This reduced reliance on the score as the main classifying feature mitigates the classifier's overall systematic uncertainties stemming from the inherent uncertainties of the score.

While composite classifiers were introduced to enhance the performance of simple LLF-based classifiers by incorporating high-level physical features of the fat jets, the reduction in systematic uncertainties has emerged as an additional benefit. Optimal utilization of the

Table 4: Background rejection at 50% signal efficiency for $GNN_{trck}$ corresponding to the datasets with $R = 0.8$ and $R = 1.2$.

| Variable | $1/\epsilon_B^c$ ($\epsilon_s^c = 50\%$) | $1/\epsilon_B^{tag}$ ($\epsilon_s^{tag} = 50\%$) |
|---|---|---|
| $R = 0.8$ | 1298 | 274 |
| $R = 1.2$ | 711 | 424 |

HLFs can boost the classifier performance while reducing dependence on the Monte Carlo generators. A comprehensive study of the performance optimization of composite classifiers while simultaneously mitigating systematic uncertainties is crucial. Nonetheless, it lies outside the boundaries of the present work, which we intend to explore in future investigations.

## 4.5  The interplay between truth-level identification, classifier efficiency, and top-tagging efficiency

The classifier's tagging efficiency ($\epsilon_S^c$) and background rejection ($1/\epsilon_B^c$), discussed in the previous section, depend on the purity of the training/testing datasets, which in turn is influenced by the truth-level tagging (TLT) criteria. The truth-level tagging/identification efficiency ($\epsilon_S^{truth}$) quantifies the fraction of hadronically decaying top quark-initiated fat jets with a given reconstruction radius ($R$) that meets the TLT criteria. Smaller $R$ values for a given $p_T$ range increase the chances of obtaining impure jets, where one or more top decay products fall outside the reconstruction cone. These impure jets are filtered out by the TLT criteria, reducing $\epsilon_S^{truth}$. For example, the $R = 0.8$ jets in the $p_T$ range $[550, 650]$ GeV, as discussed in Section 4, have a $\epsilon_S^{truth}$ of approximately 55%. Though strict TLT improves the sample purity, classifiers trained on such a high-purity sample may struggle to identify top-initiated fat jets that fall outside the TLT criteria. This is evident when we compare the background rejection rates for a given top-tagging (fourth and fifth columns) and classifier efficiency (second and third columns) in Table 2. Clearly, excellent classifier performance does not necessarily translate to higher top-tagging efficiency. Increasing $\epsilon_S^{truth}$ can help reduce this disparity. To enhance $\epsilon_S^{truth}$, one option is to relax the TLT criteria. However, doing so results in a less pure sample, leading to poorer classifier performance. Alternatively, using appropriate fat jet reconstruction radii ($R$) in different $p_T$ regions ensures that all top quark decay products remain within the reconstruction cone, thus improving $\epsilon_S^{truth}$.

In this section, we explore the impact of varying the reconstruction radius ($R$) on Truth-Level Tagging (TLT) and, consequently, on determining the classifier's tagging efficiency ($\epsilon_S^c$) and top-tagging efficiency ($\epsilon_S^{tag}$). We present our findings for the $GNN_{trck}$ classifier trained and tested with two different sets of track-based samples of signal and background fat jets falling within the $p_T$ range of $[550, 650]$ GeV. One sample used $R = 0.8$ while the other used $R = 1.2$ for fat jet reconstruction. The truth level tagging criteria are the same as discussed in Section 2. Figure 4 shows our results, with the blue curve corresponding to $R = 0.8$ jets and the red curve to $R = 1.2$ jets.

Figure 4 (left panel) shows the ROC curves when the test dataset is prepared with appropriate TLT criteria. Table 4 (second column) summarises the resulting background rejection factor corresponding to 50 % signal efficiency. The $1/\epsilon_B^c$ for the classifier trained/tested with $R = 0.8$ fat jets is close to 1300, which is 70 % higher than that of the classifier trained/tested with $R = 1.2$ fat jets. Using larger-radius jets introduces increased noise contributions from various sources. This noise can obscure the characteristic distributions of fat jets and impact the classifier's performance. While the classifier trained and tested with $R = 0.8$ jets may

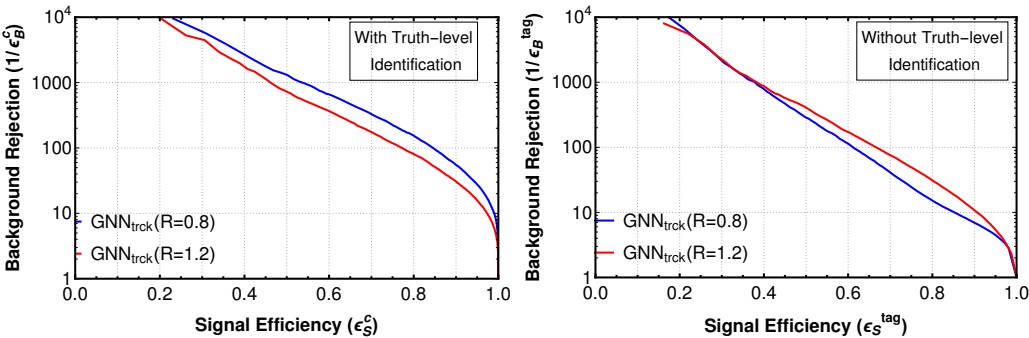

Figure 4: The ROC curves of the $GNN_{trck}$ classifier for two reconstruction radii of the fat jet, $R = 0.8$ (blue) and $R = 1.2$ (red). The left plot represents the performance for a dataset that satisfies truth-level identification criteria. No such criteria are imposed for the dataset used in the right plot.

appear impressive, as discussed previously, it does not guarantee optimal top-tagging performance. To illustrate this, we conducted tests with the same classifiers on a dataset where we did not impose any TLT criteria. The results of these tests are presented in the right panel of Figure 4 and the third column of Table 4. As anticipated, we observed a decrease in the performance of both classifiers when TLT criteria were not enforced. Interestingly, the classifier trained with $R = 1.2$ fat jets outperformed the one trained with $R = 0.8$ fat jets. The findings from this section have motivated us to explore the possibility of using different reconstruction radii ($R$) for fat jets in the six distinct $p_T$ regions that we will discuss in the following section.

## 4.6 Effect of fat jet transverse momentum on the performance of the classifier

In this section, we present the change in the classifier's performance with increasing transverse momentum of the fat jet. As discussed in Section 2, we present the performance of six classifiers, $BDT_{calo}$, $BDT_{trck}$, $CNN_{trck}$, $GNN_{trck}$, $C_{trck}B_{calo}$, and $G_{trck}B_{calo}$ in six $p_T$ ranges. We summarise our result as six plots corresponding to these $p_T$ ranges in Figure 6. Each plot presents six ROC curves, one for each classifier. For a consistent comparison of the performance of the classifiers, we present the background rejection at 50 % classifier efficiency ($\epsilon_S^c$) in Table 5. A diagrammatic representation of this result is presented in the left plot of Figure 5. Note that the fat jets in the $p_T$ range [300, 500] GeV and [500, 700] GeV have different $R$-parameters ($R = 1.2$) and hence different truth-level identification efficiency than those in the remaining $p_T$ bins where fat jets are constructed with a RR of $R = 0.8$ (see the discussion in 2). Therefore, comparing the classifier's performances for fat jets belonging to these two groups is unsuitable.

In the second and third columns of Table 5, we present the background rejection for the $BDT_{calo}$ and $BDT_{trck}$ classifiers, respectively. We see a gradual decrease in performance with increasing $p_T$. It is because the invariant mass of the QCD jets scales with its transverse momentum, and as we go higher in $p_T$, the probability of QCD jets faking as top increases. The $BDT_{trck}$ classifier performs better than the $BDT_{calo}$ because of the inclusion of additional tracking information. The fall in background rejection with $p_T$ is also smaller for $BDT_{trck}$ than $BDT_{calo}$.

The fourth and fifth columns of Table 5 represent the background rejection for the $CNN_{trck}$ and $GNN_{trck}$ classifiers. In both cases, the [300, 500] GeV $p_T$ jets have a smaller $1/\epsilon_B^c$ than the [500, 700] GeV $p_T$ jets. This is because, as demonstrated in Section 2.1, an $R$-parameter 1.2 is inefficient in capturing all the constituents of the [300, 500] GeV fat jets and reduces the performance. We see almost comparable performance for the jets in the remaining $p_T$ bins for

Table 5: Background rejection at 50 % classifier efficiency for the six transverse momentum range.

| $p_T$ [GeV] | $BDT_{calo}$ | $BDT_{trck}$ | $CNN_{trck}$ | $GNN_{trck}$ | $C_{trck}B_{calo}$ | $G_{trck}B_{calo}$ |
|---|---|---|---|---|---|---|
| 300-500 | 388 | 456 | 159 | 587 | 762 | 1413 |
| 500-700 | 136 | 276 | 184 | 765 | 455 | 1178 |
| 700-900 | 168 | 345 | 278 | 845 | 538 | 1409 |
| 900-1100 | 79 | 247 | 256 | 971 | 466 | 1175 |
| 1100-1300 | 56 | 167 | 214 | 882 | 318 | 872 |
| 1300-1500 | 39 | 127 | 217 | 877 | 273 | 850 |

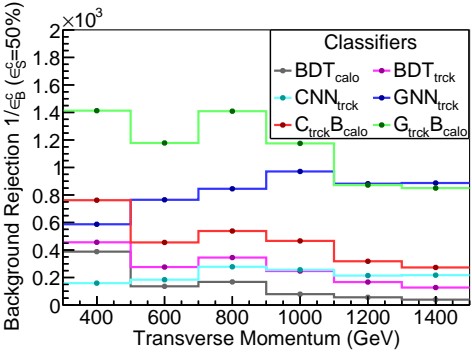 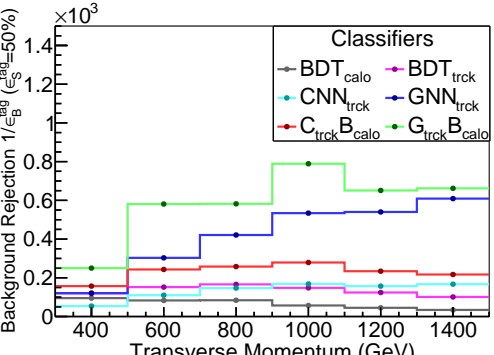

Figure 5: The variation of background rejection at 50% classifier efficiency (left) and 50% top-tagging efficiency (right) of the six classifiers corresponding to the six $p_T$ ranges considered in this paper.

Table 6: Background rejection at 50 % signal efficiency for the six transverse momentum range. Here, the testing is performed on a dataset without truth-level tagging.

| $p_T$ [GeV] | $BDT_{calo}$ | $BDT_{trck}$ | $CNN_{trck}$ | $GNN_{trck}$ | $C_{trck}B_{calo}$ | $G_{trck}B_{calo}$ |
|---|---|---|---|---|---|---|
| 300-500 | 95 | 119 | 54 | 121 | 157 | 250 |
| 500-700 | 83 | 152 | 110 | 303 | 243 | 581 |
| 700-900 | 84 | 166 | 147 | 421 | 258 | 582 |
| 900-1100 | 57 | 148 | 168 | 534 | 279 | 789 |
| 1100-1300 | 45 | 124 | 157 | 540 | 234 | 651 |
| 1300-1500 | 34 | 101 | 167 | 609 | 217 | 662 |

both classifiers. The slight reduction in performance in the case of $CNN_{trck}$ can be ascribed to the fact that with increasing $p_T$, the jet constituents get more collimated and look similar to that of a QCD jet.

Finally, in columns six and seven, we present the background rejection for the $C_{trck}B_{calo}$ and $G_{trck}B_{calo}$ classifiers. In the case of $C_{trck}B_{calo}$, we see considerable improvement compared to $CNN_{trck}$. This is because the preprocessing steps in CNN smear out the invariant mass of the fat jet. This information is restored when we combine $CNN_{trck}$ with $BDT_{calo}$ re-

Table 7: Signal efficiency corresponding to a background rejection factor $(1/\epsilon_B^{tag})$ of 1000 for the six transverse momentum range. Here, the testing is performed on a dataset without truth-level tagging.

| $p_T$ [GeV] | $BDT_{calo}$ | $BDT_{trck}$ | $CNN_{trck}$ | $GNN_{trck}$ | $C_{trck}B_{calo}$ | $G_{trck}B_{calo}$ |
|---|---|---|---|---|---|---|
| 300-500 | 22 | 25 | 16 | 27 | 30 | 35 |
| 500-700 | 20 | 27 | 22 | 38 | 32 | 42 |
| 700-900 | 20 | 28 | 26 | 40 | 32 | 43 |
| 900-1100 | 15 | 26 | 27 | 43 | 34 | 46 |
| 1100-1300 | 12 | 23 | 26 | 42 | 29 | 44 |
| 1300-1500 | 10 | 21 | 25 | 42 | 29 | 43 |

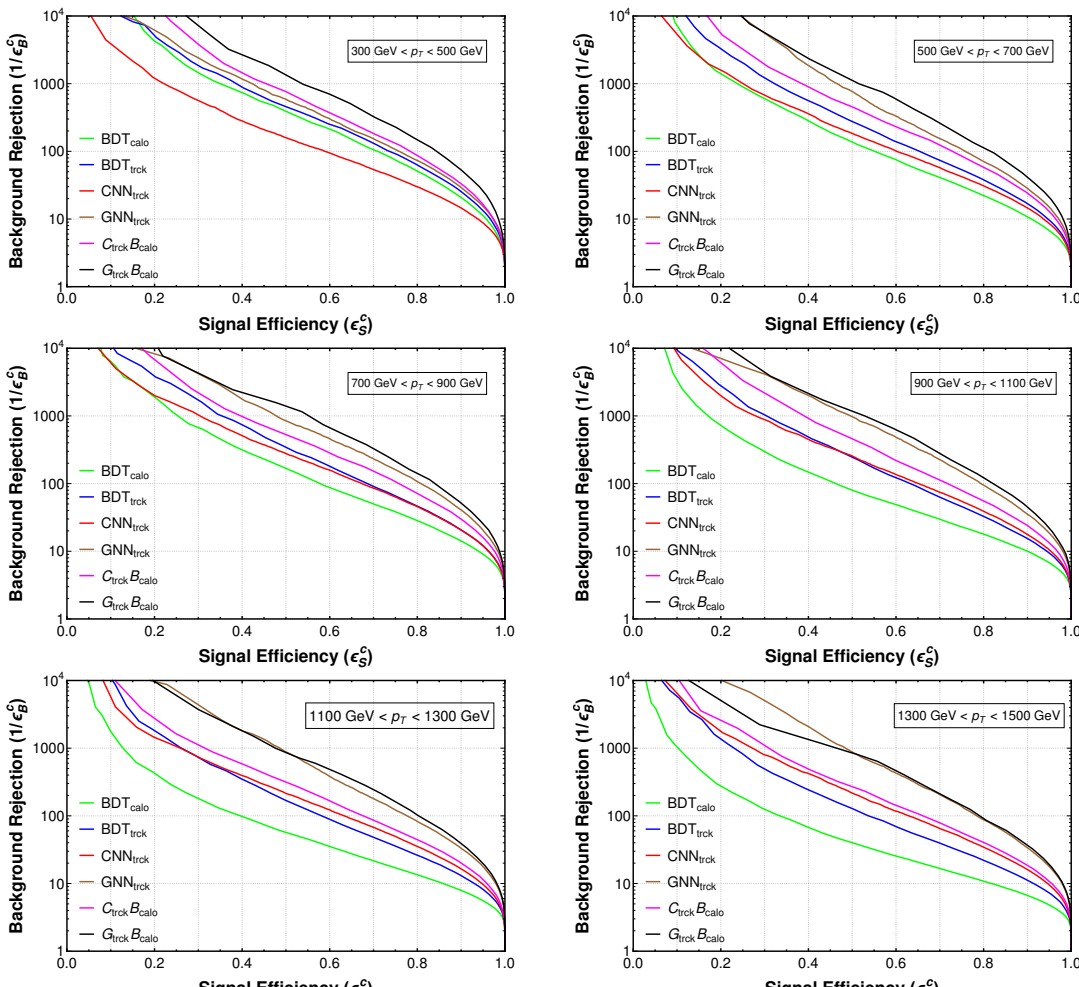

Figure 6: ROC curves for the classifiers corresponding to the six $p_T$ ranges considered in this paper.

sulting in a performance improvement (for a detailed discussion see Section 4.3). However, this improvement gradually decreases with increasing $p_T$ as the performance of the BDT decreases. The $GNN_{trck}$ classifier trained on the four-momentum data of the jet constituents can reconstruct some information about the fat jet mass. Therefore, when combined with $BDT_{calo}$,

the performance gain is not as high as in the case of $C_{trck}B_{calo}$. Here also, as we move higher in $p_T$, the performance gain gradually diminishes, and for the last two $p_T$ bins, we see almost comparable performance between $G_{trck}B_{calo}$ and $GNN_{trck}$.[20]

In Table 6, we present the background rejection at 50 % top-tagging efficiency ($\epsilon_S^{tag}$) evaluated on top samples generated without any truth-level tagging criteria. A diagrammatic representation of this result is presented in the right plot of Figure 5. As discussed in Section 4.5, the motivation for this table is to demonstrate the performance of these classifiers in a collider analysis. As expected, we see an overall degradation in performance for all classifiers. This behavior can be ascribed to improperly reconstructed fat jets in the testing sample. In Table 7, we present the top-tagging efficiency ($\epsilon_S^{tag}$) corresponding to a background rejection factor of 1000 for all the classifiers considered in our analysis. The results are evaluated using top samples generated without any truth-level tagging criteria.

## 5 Summary and outlook

We have conducted an in-depth analysis of the performance of three machine learning algorithms: the high-level feature (HLF)-based BDT, and the low-level feature (LLF)-based CNN (a miniaturized version of ResNet) and GNN (Lorentznet). Our study focused on their ability to discriminate between fat jets originating from hadronically decaying top quarks and the hadronization of light quarks and gluons. The novel findings of our research are encapsulated as follows:

**1.** A substantial portion of our study is devoted to emphasizing the significance of leveraging combined information from the calorimeter towers and tracker detectors at the LHC. We found a significant increase in the classifier's performance due to including the jet constituents' electric charge information (tracking data for charged constituents and tower data for neutral constituents) in the training and testing of the LLF-based classifiers like the CNN and GNN. Furthermore, HLF-based classifiers like BDT also exhibit performance enhancements when incorporating track-based HLFs like the number of tracks inside a jet, the $p_T$ weighted width of the tracks, the $E_T$ weighted width of the jet, etc., into the classification task. We found that the high resolution of the tracking data not only improved the classifier performance in the high-$p_T$ regions as demonstrated in Ref. [39], but the information about the distribution and composition of charged and neutral constituents of the jets coming from the tracks and towers also significantly enhance the performance of the classifiers over the whole $p_T$ range. This performance enhancement can be attributed to the fact that, according to the QCD principles and various experimental results, jets initiated by light quarks or gluons exhibit distinct differences in the distribution and composition of charged and neutral hadrons. Consequently, information about the charged and neutral constituents of a jet in the form of tracking and tower data helps identify the quark/gluon origin of sub-jets within a fat jet and hence enhances top tagging efficiency (for an in-depth discussion, please refer to section 4.2). Among the group of six simple classifiers discussed in sections 4.2 and 4.1, we found that the track-based GNN classifier ($GNN_{trck}$) consistently outperformed the others. However, it is important to note that despite their high performance, LLF-based classifiers like $GNN_{trck}$ have a significant drawback: they are heavily dependent on the jet modeling provided by the Monte Carlo simulator, such as Pythia or Herwig, which introduces substantial systematic uncertainties. We also analyzed the classifier dependence on the showering and hadronization model of the Monte-Carlo gen-

---

[20]Although $G_{trck}B_{calo}$ does not show any performance gain compared to $GNN_{trck}$, as discussed in Section 4.4, composite classifiers come with reduced dependence on Monte-Carlo generators and hence $G_{trck}B_{calo}$ results in supressed systematic uncertainty compared to $GNN_{trck}$.

erator (see section 4.4). While track-based LLF classifiers like $CNN_{trck}$ and $GNN_{trck}$ exhibited notable dependency on the Monte Carlo generators, composite classifiers (as discussed in section 4.3) demonstrated reduced dependency.

**2.** To further boost the performance of our simple HLF and LLF-based classifiers, we have developed a series of composite classifiers by stacking a BDT on top of a CNN/GNN. These composite classifiers leverage the strengths of CNN/GNN in extracting specialized observables from low-level inputs and combine them with the effectiveness of BDT in handling complex features. The result is a set of classifiers that exhibit comparable or superior performance. Please refer to section 4.3 for a comprehensive discussion. In addition to performance enhancement, the composite classifiers demonstrate reduced dependence on the jet modeling of the Monte Carlo generators (see section 4.4). The reduced Monte-Carlo generator dependency of the composite classifiers reduces the systematic uncertainties (resulting from the uncertainties in the showering and hadronisation model) to below 20%. Note that the composite classifiers do not solely rely on the event generator-sensitive scores from the LLF-based CNN/GNN classifiers. They also heavily utilize generator-insensitive HLFs such as jet mass, N-subjectness, $b$-tag, and others for the classification task. The combined use of CNN/GNN scores and other Monte Carlo generator-insensitive HLFs not only reduces overall generator dependency but also enhances their performance significantly.

**3.** We have done a comprehensive study on the interplay between truth-level identification ($\epsilon_S^{truth}$), classifier efficiency ($\epsilon_S^c$), and top-tagging efficiency ($\epsilon_S^{tag}$). Strict reconstruction and identification criteria increase the purity of the sample, simultaneously decreasing $\epsilon_S^{truth}$. A classifier trained on such pure samples is biased, and the performance cannot be efficiently generalized to new unseen data. We showed that properly selecting the reconstruction radius can improve the $\epsilon_S^{truth}$ and help mitigate this issue.

Additionally, we have demonstrated the variation in classifier performance with the transverse momentum of the fat jets. Quark and gluon jets, largely composed of QCD emissions, have an invariant mass that scales with jet $p_T$. This affects the performance of BDT classifiers where the invariant mass of the fat jet plays a key role in the classification task, and we see a considerable fall in BDT performance with increasing transverse momentum. The CNN classifier also shows a slight decrease in performance with $p_T$. This can be ascribed to the fact that the collimation of the constituents increases with transverse momentum, resulting in a top jet that resembles more with the QCD counterpart. The LorentzNet, on the other hand, is based on a Lorentz equivariant architecture and, as claimed by [76], shows almost consistent performance with $p_T$.

## Acknowledgments

The CNN model used in our analysis was discussed in the Deep Learning course 2021 at the University of Amsterdam [132]. K.G. thanks Biplob Bhattacherjee for useful discussions.

**Funding information** The simulations were partly supported by the SAMKHYA: High-Performance Computing Facility provided by the Institute of Physics, Bhubaneswar.

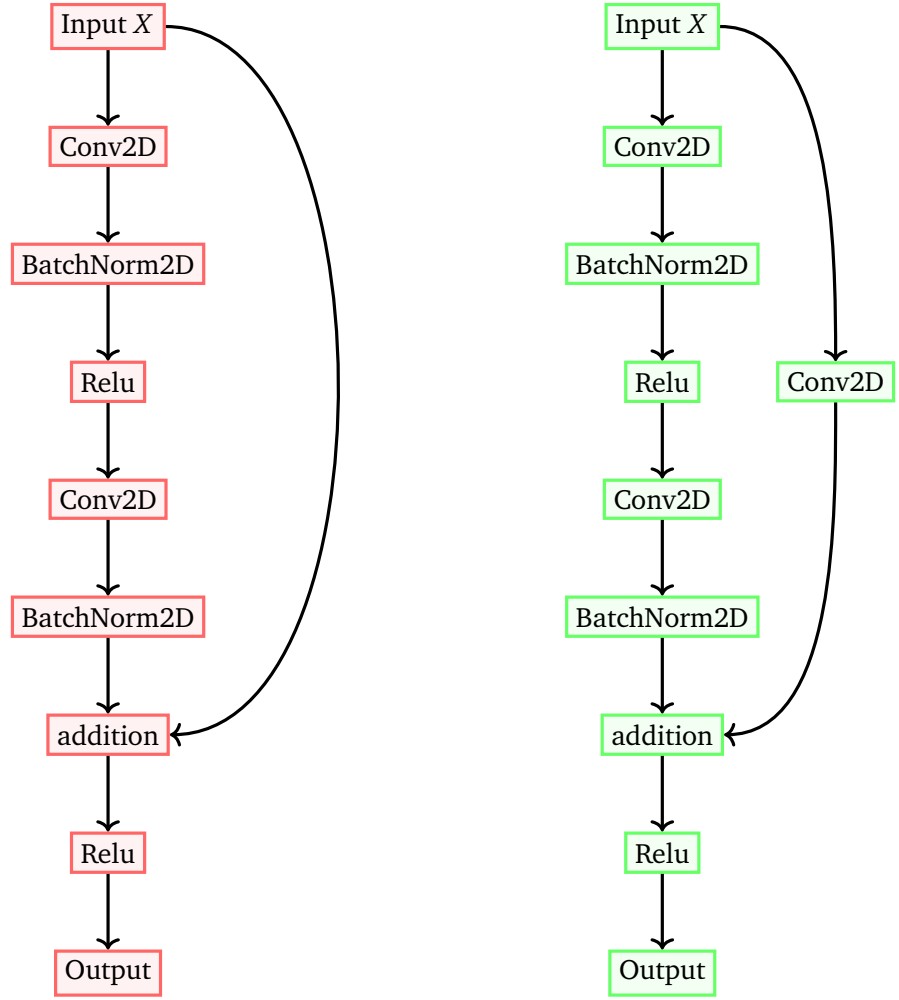

Figure 7: Diagrammatic representation of the ResNet Block without downsampling (left) and with downsampling (right).

## A The CNN model

In the left panel of Figure 7, we present a diagrammatic representation of a single ResNet block. It incorporates two convolution operations with size $3 \times 3$ filters, unit stride, and padding. Therefore, these convolution blocks cannot help us reduce the size of the input image. The number of input and output channels is also the same for the convolution operations. To reduce image size, we introduce a second convolution block, as represented in the right panel of Figure 7. Here, the first convolution layer has a stride two and unit padding and hence can reduce the height and width of the input image by half. The second convolution layer in the main network is similar to the previous ResNet block. Now, for the residual connection to work, the size of the input image must match the reduced size of the output image. We achieve this using a third convolution layer with size $1 \times 1$ filters, stride two, and no padding. In the subsequent discussion, we refer to this step as downsampling.

We present the full CNN architecture in Figure 8. First, we have an Input network that comprises the following sequence of operations:

$$Conv2D(c_{in}, c_{out}) \rightarrow BatchNorm2D \rightarrow Relu.$$

Here, the convolution layer uses $3 \times 3$ filters with unit stride and padding, $c_{in}$ and $c_{out}$ repre-

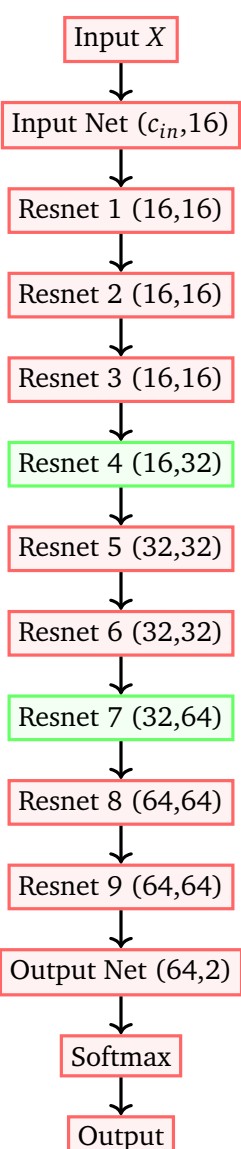

Figure 8: Schematic Diagram of the CNN architecture used in our analysis. The bracket number represents the channels in the input and output images. For the Output Net, they represent the number of nodes in the input and output layers.

sent the number of channels in the input and output images, respectively. Then, we have nine ResNet blocks stacked one after the other. The ones represented in red do not perform down-sampling of the image, whereas the green ones do. then we have the Output network, which we can represent as $AdaptiveAveragePooling(1, 1) \rightarrow Flatten \rightarrow Linear(d_{out}, 2)$, where $d_{out}$ represents the number of channels in the output image. Finally, we apply the softmax activation function to get the probability as the CNN score.

## B  Validation of our analysis

To establish confidence in our results, validating our approach (event simulation, sample preparation, etc.) by comparing our results with existing literature is crucial. In the top left

Table 8: method-specific ranking of the input features of $BDT_{calo}$.

| Variable | Ranking |
|----------|---------|
| $M$ | 0.7832 |
| $\tau_{32}$ | 0.0878 |
| b-tag | 0.0844 |
| $\tau_{2,1}$ | 0.0272 |
| $\tau_{43}$ | 0.0173 |

plot of Fig. 3, we have shown the ROC curves for $BDT_{calo}$, $CNN_{calo}$, and $GNN_{calo}$ classifiers in green, red, and blue, respectively. The $GNN_{calo}$ classifier is a slight modification of LorentzNet introduced in [76], with the difference that instead of using the mass of the constituents as node embedding, we used their charge,[21] and our training process has a smaller batch size of 16. This results in a slight difference in the classifier performance. For 50% classifier efficiency ($\epsilon_S^c$) of $GNN_{calo}$, we obtain a background rejection close to 444, while in [76], the corresponding background rejection was 498. In [99], the authors have demonstrated the performance of several classifiers on a similar dataset. $CNN_{calo}$ shows comparable performance to the CNNs presented in Ref. [99]. The similar performance observed between $GNN_{calo}$ and LorentzNet as discussed in [76], and between $CNN_{calo}$ and the different CNN-based classifiers mentioned in Ref. [99], provides validation for our methodology.

## C Correlation and ranking among variables for $BDT_{calo}$ and $BDT_{trck}$

Table 8 presents the ranking among the variables used in $BDT_{calo}$. The variables ranked higher are the ones used most frequently for node splitting. In Figure 9, we present the covariance matrix of these variables for the top jets and QCD jets.

Table 9 presents the ranking among the variables used in $BDT_{trck}$. Note that $BDT_{trck}$ uses 26 variables, and we present only the most important of them here. In Figure 10, we present the covariance matrix of the top seven highest-ranked variables for the top jets.

## D Correlation and ranking among variables for composite classifiers

This section presents the model-independent ranking and correlation among the variables used in the composite classifiers. The variable ranking for $C_{calo}B_{calo}$ and $C_{trck}B_{calo}$ are presented in Table 10. We have similar results for $C_{calo}B_{trck}$ and $C_{trck}B_{trck}$ in Table 11, for $G_{calo}B_{calo}$ and $G_{trck}B_{calo}$ in Table 12, and for $G_{calo}B_{trck}$ and $G_{trck}B_{trck}$ in 13.

The covariance matrix of $C_{calo}B_{calo}$ and $C_{trck}B_{calo}$ are presented in Figure 11, for $C_{calo}B_{trck}$ and $C_{trck}B_{trck}$ in Figure 12, for $G_{calo}B_{calo}$ and $G_{trck}B_{calo}$ in Figure 13, and for $G_{calo}B_{trck}$ and $G_{trck}B_{trck}$ in Figure 14.

---

[21]Note that the training and testing samples used in the classifiers in the left panel of Fig. 3 are generated from the tower data and hence do not have information about the jet constituent's mass or charge.

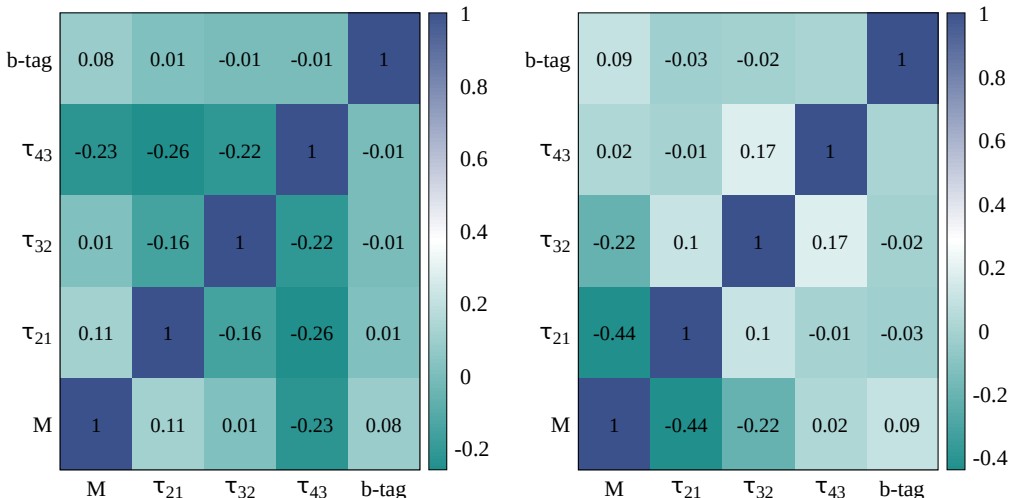

Figure 9: Correlations among the input features of $BDT_{calo}$ for the top jets (left) and the QCD jets (right).

Table 9: method-specific ranking of the input features of $BDT_{trck}$.

| Variable | Ranking |
|---|---|
| $M$ | 0.247 |
| $C_\beta(2)$ | 0.0906 |
| $\tau_{32}$ | 0.068 |
| $N_{trk}(2)$ | 0.0558 |
| $\Delta R_{1,2}$ | 0.045 |
| $\tau_{2,1}$ | 0.0415 |
| b-tag | 0.0412 |
| $w_{trk}(2)$ | 0.0385 |
| $N_{trk}(1)$ | 0.0339 |
| $C_\beta(1)$ | 0.0316 |
| $w_{trk}(1)$ | 0.0312 |
| $\Delta R_{1,3}$ | 0.0305 |
| $w_{calo}(2)$ | 0.0302 |
| $\tau_{43}$ | 0.0264 |

Table 10: method-specific ranking of the input features of $C_{calo}B_{calo}$ (left) and $C_{trck}B_{calo}$ (right).

| Variable | Ranking |
|---|---|
| $M$ | 0.3818 |
| score | 0.2685 |
| $\tau_{2,1}$ | 0.1071 |
| $\tau_{32}$ | 0.1008 |
| b-tag | 0.076 |
| $\tau_{43}$ | 0.0656 |

| Variable | Ranking |
|---|---|
| $M$ | 0.3625 |
| score | 0.309 |
| $\tau_{2,1}$ | 0.099 |
| $\tau_{32}$ | 0.09 |
| b-tag | 0.0714 |
| $\tau_{43}$ | 0.0676 |

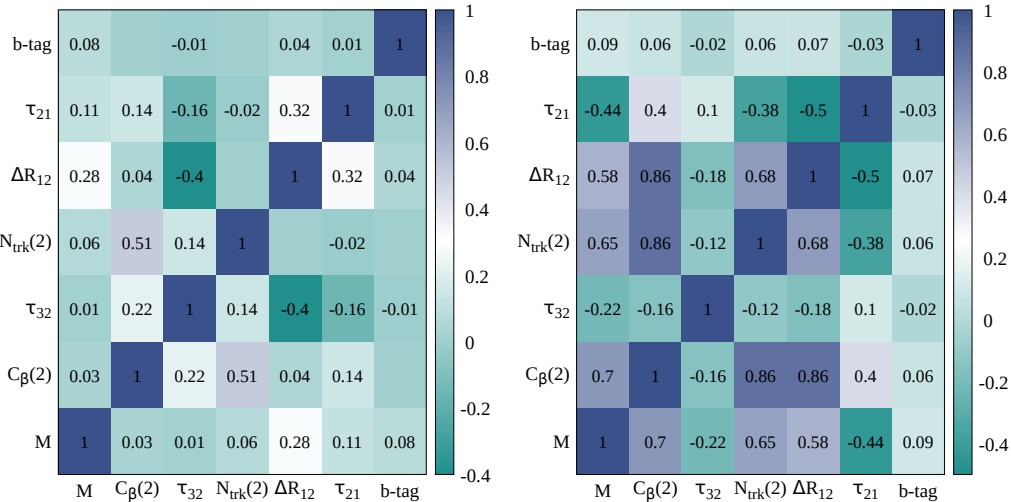

Figure 10: Correlations among the input features of $BDT_{trck}$ for the top jets (left) and the QCD jets (right).

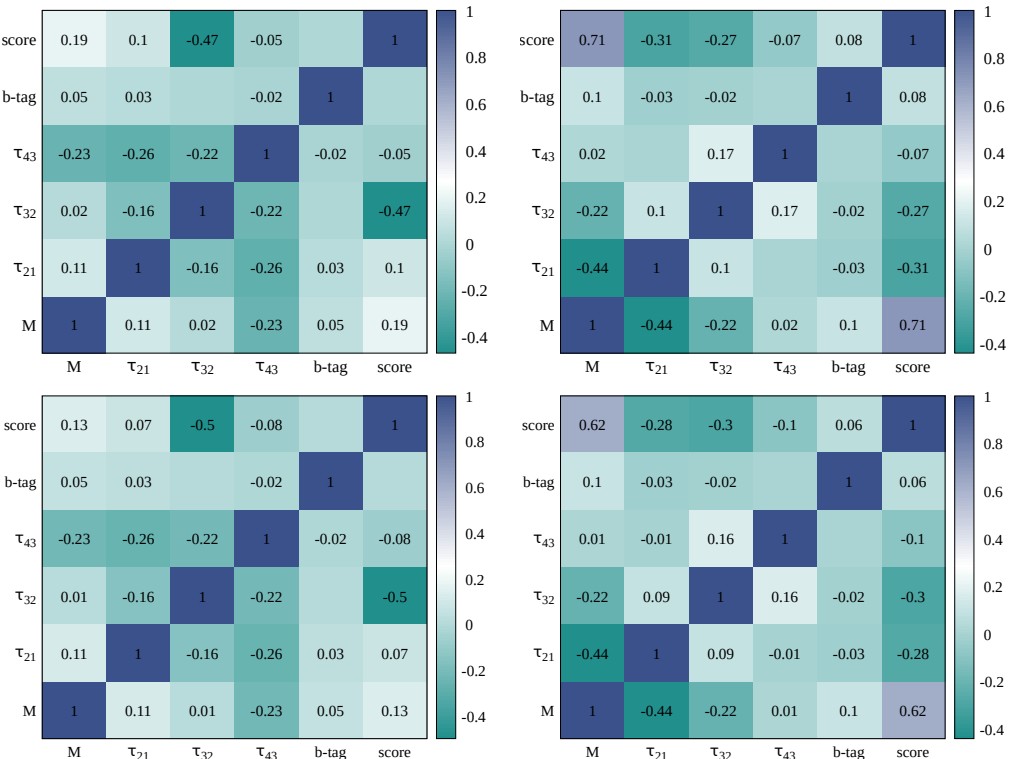

Figure 11: Correlations among the input features of $C_{calo}B_{calo}$ (top row) and $C_{trck}B_{calo}$ (bottom row) for the top jets (top left) and the QCD jets (top right).

Table 11: method-specific ranking of the input features of $C_{calo}B_{trck}$ (left) and $C_{trck}B_{trck}$ (right).

| Variable | Ranking | | Variable | Ranking |
|---|---|---|---|---|
| $M$ | 0.1811 | | $M$ | 0.1801 |
| score | 0.1328 | | score | 0.1554 |
| $w_{calo}(2)$ | 0.0814 | | $\Delta R_{1,2}$ | 0.0793 |
| $\Delta R_{1,2}$ | 0.05 | | $w_{calo}(2)$ | 0.042 |
| $N_{trk}(2)$ | 0.0493 | | b-tag | 0.03857 |
| $w_{trk}(2)$ | 0.0462 | | $\tau_{32}$ | 0.03797 |
| $\tau_{32}$ | 0.0434 | | $N_{trk}(2)$ | 0.03731 |
| b-tag | 0.03918 | | $w_{trk}(2)$ | 0.03452 |
| $w_{trk}(1)$ | 0.03639 | | $\tau_{2,1}$ | 0.03169 |
| $\tau_{2,1}$ | 0.03602 | | $C_{\beta}(1)$ | 0.03011 |

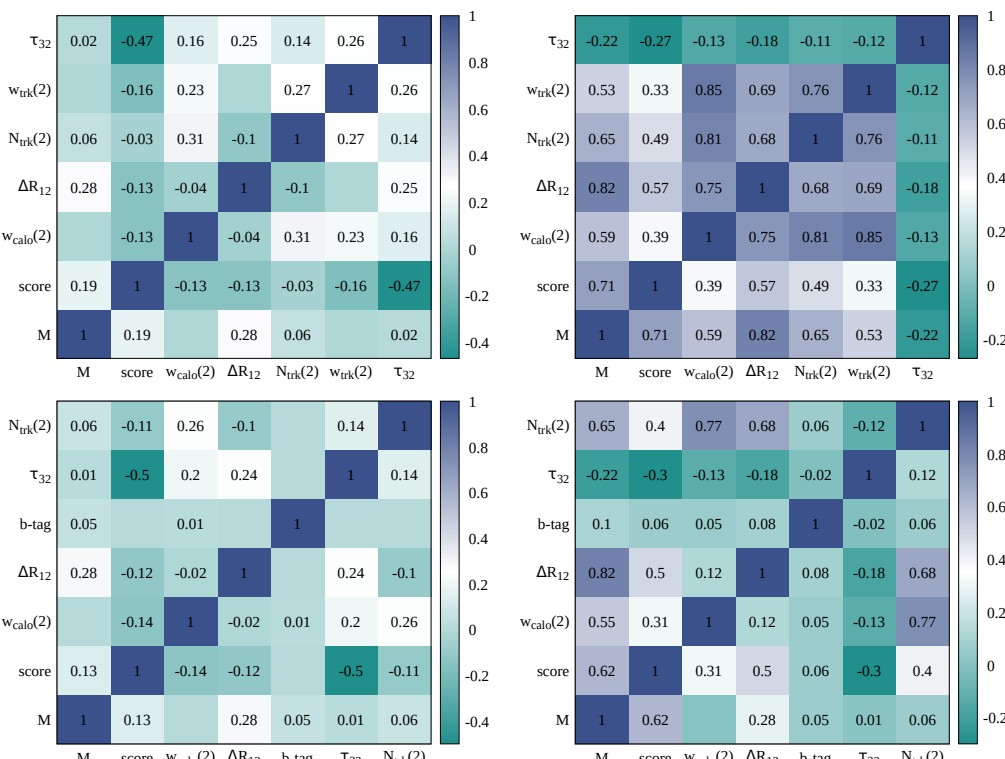

Figure 12: Correlations among the input features of $C_{calo}B_{trck}$ (top row) and $C_{trck}B_{trck}$ (bottom row) for the top jets (top left) and the QCD jets (top right).

Table 12: method-specific ranking of the input features of $G_{calo}B_{calo}$ (left) and $G_{trck}B_{calo}$ (right).

| Variable | Ranking |
|----------|---------|
| $M$ | 0.3458 |
| score | 0.322 |
| $\tau_{2,1}$ | 0.103 |
| $\tau_{32}$ | 0.094 |
| b-tag | 0.0693 |
| $\tau_{43}$ | 0.0646 |

| Variable | Ranking |
|----------|---------|
| score | 0.3517 |
| $M$ | 0.3142 |
| $\tau_{2,1}$ | 0.0968 |
| $\tau_{32}$ | 0.093 |
| b-tag | 0.075 |
| $\tau_{43}$ | 0.069 |

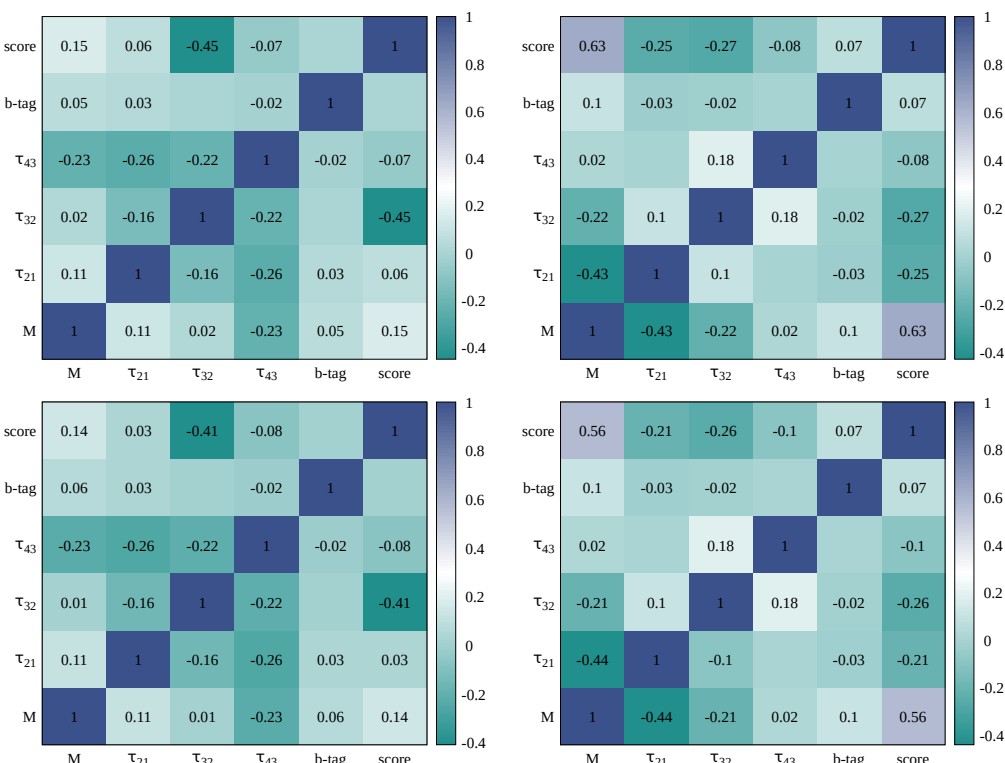

Figure 13: Correlations among the input features of $G_{calo}B_{calo}$ (top row) and $G_{trck}B_{calo}$ (bottom row) for the top jets (top left) and the QCD jets (top right).

Table 13: method-specific ranking of the input features of $G_{calo}B_{trck}$ (left) and $G_{trck}B_{trck}$ (right).

| Variable | Ranking |
|---|---|
| $M$ | 0.1696 |
| score | 0.1652 |
| $w_{calo}(2)$ | 0.067 |
| $\Delta R_{1,2}$ | 0.0568 |
| $w_{trk}(2)$ | 0.0438 |
| $N_{trk}(2)$ | 0.0407 |
| $w_{trk}(1)$ | 0.03836 |
| b-tag | 0.03775 |
| $\tau_{32}$ | 0.0369 |
| $\tau_{2,1}$ | 0.0311 |

| Variable | Ranking |
|---|---|
| score | 0.1889 |
| $M$ | 0.1624 |
| $\Delta R_{1,2}$ | 0.0755 |
| b-tag | 0.0408 |
| $w_{calo}(2)$ | 0.0354 |
| $\tau_{32}$ | 0.0351 |
| $w_{trk}(2)$ | 0.03265 |
| $\tau_{2,1}$ | 0.0321 |
| $C_{\beta}(1)$ | 0.03 |
| $w_{trk}(1)$ | 0.02931 |

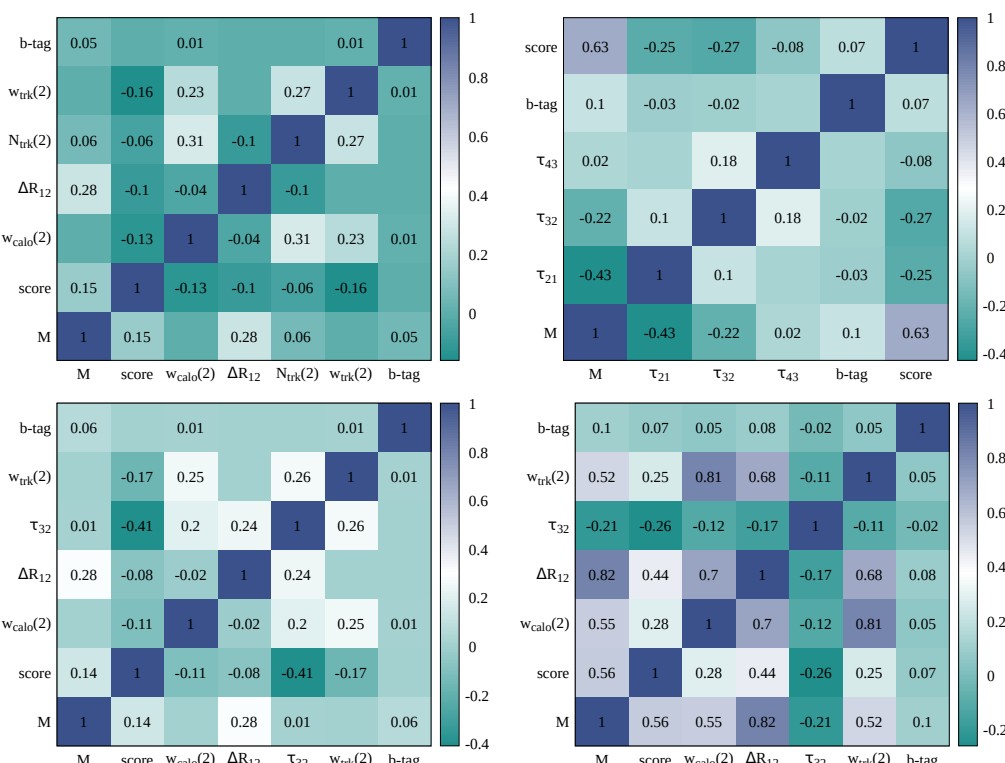

Figure 14: Correlations among the input features of $G_{calo}B_{trck}$ (top row) and $G_{trck}B_{trck}$ (bottom row) for the top jets (top left) and the QCD jets (top right).

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
