# Peer review of "ML-Based Top Taggers: Performance, Uncertainty and Impact of Tower & Tracker Data Integration"

_SciPost Physics, doi:SciPost Phys. 17, 166 (2024)_

## Round 1 · Referee Report · Anonymous (Referee 1) · 2024-5-3

Strengths
- Detailed explanations included for procedures implemented in this work.
- Comprehensive study of different types of classifiers.
- Nice summaries of main results in tables.
Weaknesses
- Important definitions are sometimes buried in the text.
- Key points are often lost in lengthy discussions.
Report
Requested changes
-
The introduction is quite lengthy and repetitive (e.g. repetition of the different architectures). Please make the introduction more concise and highlight the main points of the paper. Using bullet points to emphasize the main points and then including related details might be a good way to achieve better clarity. (Something similar to what is done in the conclusions)
-
It would be helpful for important definitions (e.g. the various kinds of efficiencies) to be explicitly defined outside the long paragraphs. This makes it easier for the reader to refer to them and ensures that these definitions are not buried in the discussion.
-
p6: In the last paragraph, you mentioned how preprocessing is meant to improve the classifier performance. However, later in Sec IVB you say that the preprocessing for the CNN smears the invariant mass distribution and results in a poorer performance. Does this mean that the CNN may be able to achieve a better performance with a different preprocessing choice?
-
p7: Please include descriptions of the preprocessing step in each row in Fig 2.
-
p16: Please enlarge the size of Fig 4.
Typos/formatting: 6. p3: 2nd paragraph line 5 - should be “fat jet” instead of “fatjet”
-
p3: 3nd paragraph - should be “Section II” instead of “section II”
-
p4: 1st paragraph - “ATLAS” instead of “Atlas”
-
p6: 2nd paragraph on right column - “second dataset” instead of “Second dataset”
-
p8: 1st paragraph - “note:” instead of “note;” (colon not semicolon)
-
p10: Footnote 14 - “parton” instead of “paron”
-
p10: Last paragraph, right before footnote 16 in text - “eps^tag_s” instead of “1/eps^tag_s”?
-
p13: 2nd paragraph - “at least” instead of “atleast”
Recommendation
Ask for minor revision
We express our gratitude to the referee for the thorough review of our manuscript. Below, we provide a summary of our responses to the comments made by the referee: The referee writes:
"The introduction is quite lengthy and repetitive (e.g. repetition of the different architectures). Please make the introduction more concise and highlight the main points of the paper. Using bullet points to emphasize the main points and then including related details might be a good way to achieve better clarity. (Something similar to what is done in the conclusions)" Our response:
Following the referee's comment, we have made several changes to the introduction section, which will be reflected in a future version of our paper. These changes include:
1. We have changed the formatting of the first paragraph on the right-hand side column of page 2 and listed the classifiers using bullet points.
2. We have removed the third and 10th-12th sentences from the second paragraph on the right-hand side column of page 2.
3. We have removed the third sentence from the first paragraph on the left-hand side column of page 3.
4. We have moved the 5th sentence from the same paragraph to footnotes.
5. Except for the last sentence, we have removed all other texts from the second paragraph on the left-hand side column of page three. We have added the last sentence to the second paragraph. These changes make the introduction section mode concise without affecting the main idea of our work. The referee writes:
"It would be helpful for important definitions (e.g. the various kinds of efficiencies) to be explicitly defined outside the long paragraphs. This makes it easier for the reader to refer to them and ensures that these definitions are not buried in the discussion.
" Our response:
We concur with the referee's recommendation and have added a paragraph in the section "CLASSIFIER PERFORMANCE" listing the definitions of the various efficiencies used in our analysis. The referee writes:
"p6: In the last paragraph, you mentioned how preprocessing is meant to improve the classifier performance. However, later in Sec IVB you say that the preprocessing for the CNN smears the invariant mass distribution and results in a poorer performance. Does this mean that the CNN may be able to achieve a better performance with a different preprocessing choice?" Our response:
The motivation behind image data preprocessing is to remove any irrelevant information from the data. It helps the machine to learn useful information and improves the training efficiency. Take the translation preprocessing step as an example. At the LHC, there is no preferred direction transverse to the beamline, and physics should be invariant to azimuthal rotation in the transverse plane. This means the distribution of a jet image at a certain azimuthal coordinate should not differ from its distribution at a different azimuthal coordinate. This motivates us to translate the jet images to be centered at the origin of the azimuthal coordinate, i.e., phi=0. Data preprocessing may come at a cost if the preprocessing step requires transformations that do not respect the symmetry of the data or lead to distortion of the information in the original data. An example of the former can be the translation of the image in the rapidity(eta) direction. We know that a translation in eta corresponds to a Lorentz boost along the beam line. If we use a variable like the energy of the calorimeter cells as pixel intensity, then a naive translation in the eta direction will change the invariant mass of the jet. However, we can choose to use variables like the transverse momentum, which are invariant under longitudinal boost, and keep the invariant mass information intact. An example of preprocessing that can alter/distort the data is the rotation step. The motivation behind the rotation preprocessing step is the observation that the radiations inside a jet are approximately symmetric about the jet axis in the eta-phi plane. The problem arises because the jet images are composed of discretized pixels, and any rotation other than a factor of "pi/2" cannot be exact. To perform rotation by arbitrary angles, we usually follow some interpolation methods, and this step can smear out the invariant mass information of a jet. As far as we know, there is no well-defined solution to this problem, though some literature recommends removing this preprocessing step completely (see, for instance, "arXiv:1701.08784"). Note that it is also an option to not use any preprocessing steps in the analysis. However, this requires a large data sample so that the classifier can effectively learn on its own which information is actually important for the classification task and which is redundant. We do not follow this procedure because of the huge computational cost involved. As far as top tagging is concerned, most analyses in the literature follow the four-stage preprocessing involving translation, rotation, reflection, and normalization. For our analysis, we followed the above preprocessing choice to set up a framework that is easy to relate to other similar analyses in the literature. The referee writes:
"p7: Please include descriptions of the preprocessing step in each row in Fig 2." Our response:
We have included a brief description of the preprocessing steps corresponding to each row of Fig 2 in the figure caption. The details of these preprocessing steps are in the text in the last paragraph of the subsection titled CNN. The referee writes:
"p16: Please enlarge the size of Fig 4." Our response:
We have made the figure larger in the manuscript. In points 6 through 13, the referee has pointed out a few typos/formatting errors in the manuscript. We would like to thank the referee for pointing out these typos. We have corrected these errors in the manuscript.

Author: Rameswar Sahu on 2024-09-18 [id 4786]
(in reply to Report 3 on 2024-05-30)We thank the referee for reviewing the manuscript. Surely, the referees' comments helped us prepare a better manuscript.
Comment/criticism #1: When discussing the jet images for the CNN training, in the example you rescale the images with 16x16 (arising from the resolution of the detector) up 64x64 pixels. Can you elaborate here? Why not use a CNN for the 16x16 image?
Our response: We recon with the referee's observation that $16 \times 16$ images could also be used for training the single-layered CNN ($CNN_{calo}$). However, for our analysis, we aimed to compare the performance of the single-layered CNN with the two-layered CNN ($CNN_{trck}$). For the latter case, the second layer of the jet image was constructed from the tracker hits, and the better resolution of the tracker allowed us to use images with dimensions $64\times 64$ (a $64 \times 64$ image contained more information than a $16\times 16$ image and can help the CNN learn better). Since both layers of the image have to be of the same size, for $CNN_{trck}$, we are also required to design the first layer (which utilizes the calorimeter energy deposits as pixel intensities) as a $64\times 64$ image. Since the images in $CNN_{calo}$ only use the calorimeter energy deposits, we also made the single-layered images $64 \times 64$ dimensional to better compare its results with that of $CNN_{trck}$.
Comment/criticism #2: For the GNN / CNN training, is there a particular reason why you train the model for 35 epochs with such small batch-sizes (16 and 32)? Why not train for more epochs with larger batch-size? For the CNN a small batch-size can also degrade the results when using BatchNorm layers.
Our response: The choice of epoch number is motivated by earlier work arXiv:2201.08187 [hep-ph], where the original LorentzNet model was introduced. In our analysis, we experimented with varying the number of training epochs for the CNN/GNN models and found that with 35 epochs, the CNN/GNN can provide a satisfactory performance. As the referee correctly observed, using a larger batch size can indeed enhance CNN performance. However, our decision to use smaller batch sizes was necessitated by the constraints of our computational resources. Handling larger data batches demands more memory than our current computational infrastructure can support.
Comment/criticism #3: How exactly is the b-tagging performed for the BDT? Notice that the b-tagging efficiency should degrade substantially for higher jet pT's (above 200 GeV) when using vanilla b-taggers. Are you including this in your b-tagging?
Our response: As the referee has rightly pointed out, the b-tagging efficiency degrades with increasing jet $p_T$. For our analysis, we have used the default b-tagging algorithm of Delphes, where a $p_T$ dependent b-tagging efficiency based on ATL-PHYS-PUB-2015-022 is implemented.
Regarding the impact of the b-tagging variable on the performance of the BDT classifiers, we found that its contribution is not particularly significant. This is primarily due to the presence of other dominant discriminating variables, such as mass (M) and $\tau_{32}$ for the $BDT_{calo}$ classifier (see Table VIII), and mass (M), $C_{\beta}$, $\tau_{32}$, $N_{trk}$, etc., for the $BDT_{trk}$ classifier (see Table IX). As a result, while the b-tagging efficiency does degrade at higher jet $p_T$, its effect on the overall performance of the classifiers remains limited.
Comment/criticism #4: None of the ROC curves have uncertainties. Have you checked if your results change substantially when training the models multiple times with different random seeds?
Our response: We have not included uncertainties in the ROC curves due to the large number of models studied in our analysis. As the referee has correctly pointed out, calculating such uncertainty would require training the models multiple times with different seeds. The LorentzNet and ResNet architectures used in our study are computationally intensive, with each training run taking over a day on our available hardware. Consequently, generating uncertainty estimates for the ROC curves of both simple and composite classifiers would have been exceedingly time-consuming.
Nevertheless, the LorentzNet architecture and dataset in our study closely resemble those in the original analysis arXiv:2201.08187 [hep-ph], where the authors demonstrated that predictions remained stable across different random seeds. During our analysis, we verified this behavior for the simple GNN classifiers using fat jets in the 300-500 GeV $p_T$ range. We also confirmed that the results of our simple CNN classifiers were similarly stable across different seeds. However, since we have not done similar studies for the other datasets and models, we have not included these results in the manuscript.
Comment/criticism #5: All of the composite classifiers are simple BDTs that use a CNN or GNN score as one of its HLFs. Wouldn't the reduction in MC generator uncertainty be somehow related to the BDT rank of the score feature? Is there any simple way of quantifying this reduction?
Our response: We agree with the referee's observation that the rank of the score feature might be related to the reduction of the MC generator uncertainty. While this idea is intriguing, establishing such a connection is not straightforward. It would require training a classifier (BDT/ANN) multiple times, adjusting the importance of the score feature, and studying how this affects the MC generator uncertainty. Such an analysis is beyond the scope of our present work. However, we are planning a future analysis solely focused on handling this uncertainty, and there, we plan to address this issue in detail.
Comment/criticism #6: Any reason why you did not stack CNN + GNN + BDT?
Our response: A composite model with stacked BDT, CNN, and GNN will be highly complex, and we do not expect a significant performance boost from such a combination.
Table III of our paper shows that the $GNN_{trk}$ model performs exceptionally well when given complete information, such as the exact mass of the jet constituents. Our analysis acknowledges that it is currently not feasible at the LHC to measure the constituents' mass with full accuracy. To account for this uncertainty, we have treated all jet constituents as massless in our study. As a result, the graphs used in our analysis lack complete information on the fat jet.
When we stack BDT on top of the GNN, the high-level features of BDT provide this missing information. This is evident in Table III, where both $GNN_{trk}$ (with complete information) and $G_{trk}B_{trk}$ (where the graphs exclude constituents mass information) achieve nearly comparable performance. This compels us to believe that adding CNN to the combination of BDT and GNN is unlikely to yield any meaningful performance improvement.
General comments #1: Shorten introduction substantially by removing information that is not directly relevant for this work. The conclusion should be used as a template.
Our response: In line with the referee's suggestion, we have significantly shortened the introduction by removing content that is not directly pertinent to the core focus of our work.
General comments #2: In the Dataset section I would recommend putting the details of the MC simulations into an appendix.
Our response: We concur with the referee's observation that the details of the MC simulations are a bit technical, and many readers may find our results more appealing than the details of the simulation. However, we feel understanding the dataset can help the readers gain better insight into our results. Especially the details of the Delphes classes from which the jet constituents are extracted are important in understanding why the track-based classifiers perform better than the tower-based ones. It also clarifies which constituents of the fat jets are getting zero mass in the dataset used for the GNN-based classifiers. Therefore, keeping this information in the main body of the paper will be more informative for the readers.
General comments #3: There is no need to provide details on LorentzNet in the Model section. It is enough to cite the original reference. Same with the CNN sub-section.
Our response: We have removed details regarding the LorentzNet model and referred the readers to the original paper for details.
General comments #4: Move validation sub-sections in sec IV add to an appendix
Our response: We have moved the validation section to the appendix.
General comments #5: "Centralize" all results in a single section (currently results are scattered in two sections), and try to just keep the more relevant results in the bulk and the rest in an appendix
Our response: We have merged our findings into a single section, keeping only the relevant information in the main body of our paper.

---

## Round 1 · Referee Report · Anonymous (Referee 2) · 2024-5-15

Strengths
2 - Exploration of composite classifiers as a way to include both high and low level features in classification.
3 - Nice observation on the effect on systemic uncertainties when using the composite classifiers on top of deep neural nets.
Weaknesses
2 - The most significant result appears to be the simultaneous reduction in systematic error and improvement in tagging performance that comes from using composite classifiers, but this is not studied in so much detail. The authors say that this further work will be done in the future.
Report
With the requested changes below, I would recommend this paper for publication in SciPost Core.
Requested changes
1 - The introduction should be shortened to include the main points only.
2 - The results sections can also be shortened/combined to avoid too much repetition of the results and discussion.
3 - Can you comment on the importance of the b-tag information when comparing the performance of the BDT, CNN/GNN, and composite classifiers?
Recommendation
Accept in alternative Journal (see Report)
We extend our thanks to the referee for their thorough examination of our manuscript. Below, we summarize our responses to the referee's comments:
We concur with the referee's first two points and plan to implement them in a future version of our paper. As per the third point,
The referee writes:
"3 - Can you comment on the importance of the b-tag information when comparing the performance of the BDT, CNN/GNN, and composite classifiers?"
Our response:
B-tagging information plays a significant role in separating top jets from QCD jets. A quantitative estimate of its importance can be inferred from the method-specific ranking of input features for the calorimeter-based BDT classifier shown in Table VIII of our paper. However, with the inclusion of track-based features, the importance of this variable decreases as the tracking information is more important in the discrimination of quark jets (top decay product) from gluon jets (a major component of the QCD background jets) (more details provided in the paper). Similarly, in the case of the composite classifiers, due to the presence of other important features, the importance of the b-tagging information is masked.
As for the CNN and GNN-based classifiers, as they are trained directly with the jet-constituents four-momentum as input, they do not have direct access to the b-tagging information. However, one may argue that since the classifiers already have the information of all constituents, they can use this information to gain knowledge about the b-jet inside the top-jet. But this does not work. As shown in the correlation plots in Figure 11 of our paper, the b-tagging information has little to no correlation with the CNN/GNN score. This indicates that the CNN/GNN score and b-tag variable carry independent information. This behavior of not recognizing some important high-level features of the input data (in this case, the b-tagging information) is a well-known problem with low-level feature-based classifiers like CNN/GNN. As far as we know, three different strategies exist in the literature that can address this problem. One of these methods is the construction of composite classifiers, as demonstrated in our paper. The second strategy is to introduce the high-level features as additional global variables in CNN and GNNs. Finally, the third method is a more recent one and advocates using an attention mechanism that forces the CNN/GNN to learn more about these high-level features (See arXiv:2403.11826 for more on the second and third strategies).

---

## Round 1 · Referee Report · Anonymous (Referee 3) · 2024-5-30

Strengths
-
The authors demonstrate that enhancing high-level BDT taggers with the classification scores of GNN/CNN classifiers trained on low-level features improves the tagging performance while mitigating systematic uncertainties from MC generators.
-
They show that adding HL tracking information to the taggers also improves performance.
-
The authors give thorough explanations and lots of details on the procedures they used.
Weaknesses
-
The way the manuscript is structured makes it difficult to read.
-
The authors do not discuss how the MC generator uncertainties present in coming from the low-level features propagate into the composite BDTs.
-
No uncertainties for the ROC curves are provided.
Report
Overall, the work in this paper is interesting and novel. However, the paper is very difficult to read. Information that should go together is sometimes scattered in different places, and in some parts, information is repeated, resulting in an overload that is hard to follow.
I suggest the authors carefully restructure the paper to increase clarity. I believe the content of the paper merits publication, but not in its current format. However, if the authors address this issue and also answer my questions below, I would be happy to recommend it for publication.
-
When discussing the jet images for the CNN training, in the example you rescale the images with 16x16 (arising from the resolution of the detector) up 64x64 pixels. Can you elaborate here? Why not use a CNN for the 16x16 image?
-
For the GNN / CNN training, is there a particular reason why you train the model for 32 epochs with such small batch-sizes (16 and 32)? Why not train for more epochs with larger batch-size? For the CNN a small batch-size can also degrade the results when using BatchNorm layers.
-
How exactly is the b-tagging performed for the BDT? Notice that the b-tagging efficiency should degrade substantially for higher jet pT's (above 200 GeV) when using vanilla b-taggers. Are you including this in your b-tagging?
-
None of the ROC curves have uncertainties. Have you checked if your results change substantially when training the models multiple times with different random seeds?
-
All of the composite classifiers are simple BDTs that use a CNN or GNN score as one of its HLFs. Wouldn't the reduction in MC generator uncertainty be somehow related to the BDT rank of the score feature? Is there any simple way of quantifying this reduction?
-
Any reason why you did not stack CNN + GNN + BDT?
Requested changes
1- Shorten introduction substantially by removing information that is not directly relevant for this work. The conclusion should be used as a template.
2- In the Dataset section I would recommend putting the details of the MC simulations into an appendix.
3 - There is no need to provide details on LorentzNet in the Model section. It is enough to cite the original reference. Same with the CNN sub-section.
4 - Move validation sub-sections in sec IV add to an appendix
5 - "Centralize" all results in a single section (currently results are scattered in two sections), and try to just keep the more relevant results in the bulk and the rest in an appendix
Recommendation
Ask for major revision

---

## Round 2 · Referee Report · Anonymous (Referee 1) · 2024-10-9

Report

The paper is much more clearly written now. As requested in my previous report, important definitions are now explicitly stated which make them easier to refer to while reading the paper. The authors have sufficiently addressed the main concerns presented in my previous report and I find the revised version acceptable for publication.

Recommendation

Publish (meets expectations and criteria for this Journal)

---

## Round 2 · Referee Report · Anonymous (Referee 2) · 2024-11-3

Report

The authors have improved the manuscript a lot, particularly the introduction and conclusions. I do think this paper now meets the criteria to be published in SciPost Physics Core. While the paper and the results are valuable, I do not think the techniques studied are new or novel enough to warrant publication in SciPost Physics.

Recommendation

Accept in alternative Journal (see Report)

---

## Round 2 · Referee Report · Anonymous (Referee 3) · 2024-11-8

Report

The authors have addressed all of the questions and concerns raised in my previous report, resulting in a substantial improvement in the quality of the manuscript. I am satisfied with the revisions made and recommend this paper for publication.

Recommendation

Publish (meets expectations and criteria for this Journal)

---

## Round 2 · List of Changes

1) We have significantly shortened the introduction by removing content that is not directly pertinent to the core focus of our work. 2) We have removed details regarding the LorentzNet model in the model section of our paper and referred the readers to the original paper for details. 3) We have moved the validation section to the appendix. 4) We have merged our findings into a single section, keeping only the relevant information in the main body of our paper. 5) We have significantly shortened the result section by keeping only the relevant results.

---

## Editorial Decision

published